



# Seasonal variation in size-resolved particle deposition and the effect of environmental conditions on dry deposition in a pine forest

Erin K. Boedicker[a], Holly M. DeBolt[a], Ryan Fulgam[a,*], Ethan W. Emerson[a,b], and Delphine K.

Farmer[a]

[a]Department of Chemistry, Colorado State University, Fort Collins, CO, USA; [b]Handix Scientific, Fort Collins,

CO, USA

* Now at Environmental Protection Agency, Durham, North Carolina, USA

*Correspondence to*: Delphine K. Farmer Delphine.Farmer@colostate.edu Department of Chemistry, Colorado

State University, 1301 Center Ave, Fort Collins, CO, 80523, USA.

**Abstract**

Dry deposition is a fundamental process that removes particles from the atmosphere, and therefore directly
controls their lifetime and total impact on air quality and radiative forcing. The processes influencing dry
deposition are poorly constrained in models. Seasonal changes in dry deposition remain uncertain due to the
lack of observations over multiple seasons. We present measurements of size-resolved sub-micron particle
deposition from a flux study that surveyed all four major seasons. Particle concentrations and therefore fluxes
were highest in the summer and lowest in the winter. Size-dependent deposition velocities in all seasons were
consistent with previously observed trends, however, our observations show a $130 \pm 60\%$ increase in wintertime
deposition velocity compared to the summer, which is not currently captured in size-resolved deposition
models. We explore the influence of scalar gradients and changes in environmental conditions as possible
drivers of this increase. We find that phoretic effects, such as thermophoresis, and the addition of snow to the
canopy had negligible impacts on our canopy level measurements. While turbophoresis impacted the observed
seasonal changes in size-resolved particle deposition velocity, it did not fully explain the observed differences
between the summer and winter. We suggest that the increase in deposition velocity is instead caused by
changes to the leaf-level conditions and physiology during the wintertime, which increase interception of
particles.

## 1  Introduction

Particles impact the quality of the air and radiative balance of Earth's atmosphere as a function of their size,
chemical composition, and lifetime. The lifetime of particles is controlled by their rate of removal. The two removal
pathways for particles from the atmosphere are wet deposition, which is the scavenging of particles and subsequent
removal by precipitation, and dry deposition, which is the removal of particles through interactions with terrestrial
and aquatic surfaces. Dry deposition of particles in the accumulation mode is currently the largest source of
uncertainty in global models in the prediction of concentrations of cloud condensation nucli and the prediction of
particle impacts on cloud albedo (Carslaw et al., 2013; Lee et al., 2012, 2013). Uncertainty in dry deposition
removal rates stem from both inaccuracy in current theoretical parameterizations and the limited spatial and
temporal coverage of measurements, especially over certain terrestrial surfaces during seasons other than
summertime. Understanding the removal of particles through dry deposition is critical for constraining particle
lifetime and therefore total impact in the atmosphere. While there have been major strides in quantifying and



understanding dry deposition, large gaps in our knowledge of underlying deposition mechanisms persist (Farmer et al., 2021; Saylor et al., 2019; Hicks et al., 2016; Pryor et al., 2008).

Several terms can be used to describe the biosphere-atmosphere exchange of particles. Particle flux describes the net exchange of particles and is strongly influenced by changes in particle concentration and size distribution in an
environment. Environments exhibit seasonal changes in concentration and size distribution that can influence particle flux; this is easily observed in urban environments where local particle sources persist. During the winter in urban regions particle concentration and therefore flux often increase due to an increased use of heating systems (Casquero-Vera et al., 2022). In contrast, rural environments often see increases in particle flux during the summer due to the dominance of biogenic secondary organic aerosol. In the same way, size distribution changes can
influence particle flux. For these reasons particle flux is ideal for identifying sources of particles within an environment. Exchange velocity ($V_{ex}$) is derived from the flux and is independent of the particle concentration, describing the vertical directional movement of particles in an environment. Both flux and $V_{ex}$ follow the same sign convention where positive indicates upward movement, or emission, and negative indicates downward movement, or deposition. Deposition velocity ($V_{dep}$) describes the portion of exchange velocities that have a downward direction
– using a positive sign for velocities directed downward towards biosphere surfaces– and defines how quickly particles are collected by the surface. Since these terms are independent of concentration and size distribution changes, their behavior should remain unchanged unless there are shifts in the underlying mechanisms driving these velocities.

The extent to which the mechanisms behind dry deposition velocity over terrestrial ecosystems vary with ecosystem
properties or other seasonally varying parameters remains poorly constrained by observations. Few studies have presented either bulk aerosol deposition velocity (e.g. Zhai et al. 2019; Ahlm et al. 2010; Rannik et al. 2009; Suni et al. 2003) or size resolved deposition measurements across seasons (e.g. Petroff et al. 2018; Mammarella et al. 2011; Gallagher et al. 1992).  These studies typically observe seasonal differences in both particle flux and deposition velocity and have raised several hypotheses regarding seasonality of particle dry deposition. In their study of cloud
droplet deposition (3 – 31 µm), Gallagher et al. (1992) observed a reduction in deposition velocity during wintertime snow cover that they attributed to the addition of snow to the canopy thereby reducing overall surface roughness. In contrast, Mammarella et al. (2011) observed an increase in deposition velocity in the winter, while the fall and summer periods had the lowest deposition velocities. Mammerella et al. suggested wintertime enhancements in ultrafine (0.020 – 0.065 µm) particle deposition velocities were due to thermophoretic effects, and changes to other
particle sizes were due to either turbophoretic effects or changes in the concentration of various modes in the seasonal distributions. Thermophoresis occurs when thermal gradients drive particles towards the collecting surface. When the air is warmer than the collecting surface the increased energy in the space above the surface drives collisions and interactions that propel particles away from each other and towards areas with smaller energy reservoirs, which in this case is the colder collecting surface. Turbophoresis is the movement of particles by
gradients in turbulence. For this mechanism, increased turbulence in areas above the canopy and collecting surface drive particle collisions and interactions which in turn move particles away from the areas of high turbulence and towards regions of low turbulence. In this way both processes enhance the interaction time between the particle and the surface, therefore increasing the deposition velocity. Phoretic effects describe the effects of all scalar gradients, including thermophoresis and turbophoresis, and have been explored as contributors to deposition over snow and ice
surfaces (Petroff and Zhang, 2010; Tammet et al., 2001). Turbophoresis in particular has been considered as a contributor to particle deposition in forests where trees create significant gradients in turbulence between the top of the canopy and the forest floor (Katul et al., 2010; Feng, 2008). Another seasonal factor that could impact dry deposition is the condition of the forest canopy or vegetation structure. Particle uptake by vegetation is a significant contributor to dry deposition (Pryor et al., 2017), and plant morphology and physiology impacts particle deposition
(He et al., 2020; Räsänen et al., 2012, 2013). Changes to the canopy structures affect eddy penetration which could also impact turbophoresis. Seasonal changes in leaf-level conditions could therefore have significant impacts on dry deposition. The relative influence of each of these proposed mechanisms has not been critically evaluated, due to the limitations of available measurements, but they have been incorporated in some particle deposition models.

The objective of this work was to investigate seasonal variation in particle concentration and fluxes over a temperate
pine forest from the Seasonal Particles in Forests Flux studY (SPiFFY) in 2016. We present general trends in



meteorology, particle concentration and distribution, and particle flux across the four seasons: winter, spring, summer, and fall. Additionally, we present seasonal trends in total and size-resolved particle deposition velocity compared to current resistance model parameters. This work explores the drivers of seasonal variations in particle deposition and presents new evidence for the addition of previously neglected mechanisms in dry deposition
modules as well as the inclusion of seasonally specific constants.

## 2   Methods

### 2.1   Site Description

The Seasonal Particles in Forests Flux studY (SPiFFY) was conducted at Manitou Experimental Forest Observatory (MEFO) located within Manitou Experimental Forest in central Colorado, USA (39.1006°N, 105.0942°W). Four measurement campaigns were performed between 2015 and 2016, each representative of one of the major seasons: winter (February 1 – March 1, 2016), spring (April 15−May 15, 2016), summer (July 14−September 16, 2016), and fall (October 1−November 1, 2016). Manitou Experimental Forest is approximately 6760 ha of ponderosa pine, Douglas fir, mixed conifer, and aspen vegetation with an average canopy height of 16 m. Manitou Experimental
Forest elevation ranges from 2280 – 2840 m above sea level. The MEFO tower site is described in detail by Ortega et al. (2014). Measurements were made at the 30 m walk-up tower at MEFO, with instrumentation installed in an exterior trailer at the base of the tower. The footprint of the tower was dominated by Ponderosa pine trees reaching 16 m in height.

Fulgham et al. (2019) summarized the meteorology of the site across the four seasons during SPiFFY, which is
presented in **Figure 1** and **Table 1**. The friction velocity ($u*$) and sensible heat flux ($H$) followed the conventional diel cycles associated with increases in solar heating during the day, enhancing turbulence, and decreases at night (**Figure 1**). This trend is observed in all four seasons, with the summer and fall having the strongest increase in sensible heat during the day. Friction velocities were comparable across seasons; however, the winter did exhibit higher turbulence during nighttime periods than other seasons. Further details of the site set up and these trends can
be found in Supplemental **Figure S1 & S2**.

### 2.2   Instrumentation

Size-resolved particle concentrations were measured using an Ultra-High Sensitivity Aerosol Spectrometer (UHSAS; DMT Inc., Longmont, CO; Cai et al. 2008). The UHSAS had a 10 Hz time resolution and counted
particles in 99 size bins (0.06 – 1 µm). The data was re-binned during analysis to 10 size bins, and data from 0.06 – 0.089 µm was not included in the analysis due to the presence of noise that caused irregular and severe fluctuations in the signal of that size range. Calibrations were run at the beginning of each deployment using NIST standard polystyrene latex spheres (60, 150, 300, 600, and 900 nm mobility diameter). System zeros were run using a HEPA filter and switch installed at the front of the main line. The inlet set-up for the measurements consisted of a 28.98 m
(ID: 7.14 mm, OD: 9.53 mm) copper line installed 25 m above ground level on the tower and was co-located with the sonic anemometer. A cone with metal mesh to keep out bugs and debris was affixed to the front of the inlet. Flow through the main line was maintained at ~20 L/min (Re ≈ 3900; residence time of 3.5 s in the main line) using a backing pump and mass flow controller. Inside the trailer, a 3.05 m line reduced to ID: 4.83 mm (OD: 6.35 mm), and the UHSAS sampled off the main line at 60 mL/min through its internal conductive silicone tubing (OD: 3.2
mm) (**Figure S1**). Particle loss in the main line calculated using the method described by Von der Weiden et al. (2009) was determined to be negligible (<5%).

Two different sonic anemometers were used during the campaign to measure three-dimensional wind speed and temperature. For the winter, spring, and summer periods (February 1 – August 5, 2016), we used a CSAT3 sonic anemometer (Campbell Scientific, Logan, UT). Inlet offsets from the CSAT were $x$ = 20 cm, $y$ = 40 cm, $z$ = 10 cm.
The fall measurements (October 1−November 1, 2016) used a SATI-series K-style sonic anemometer (Applied Technologies Inc., Boulder, CO). Inlet offsets from the SATI were $x$ = 0 cm, $y$ = 0 cm, $z$ = 60. Both anemometers were set to record data at 10 Hz time resolution. Data from the CSAT3/SATI and UHSAS were logged on separate computers and the timestamps were synched to an online time server. Any observed clock drifts in the data were handled in post processing.






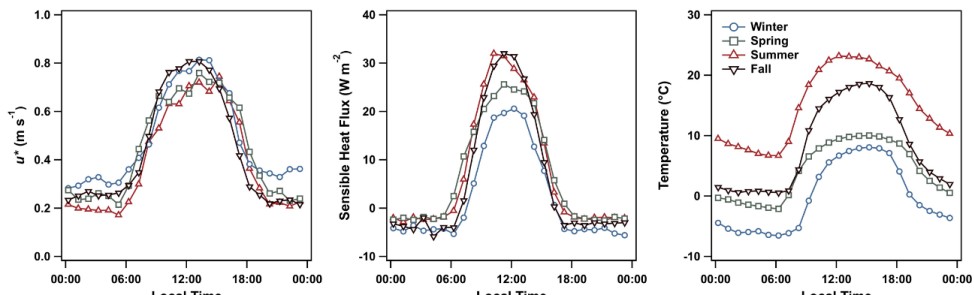

**Figure 1**: Average diel cycle for friction velocity ($u^*$), sensible heat flux ($H$), and air temperature of each season. Each point represents an hourly median value.

**Table 1**: Seasonal environmental daytime conditions, originally published by Fulgham et al. (2019).

|  | Friction Velocity (m/s) | Wind Speed (m/s) | Temperature (°C) | Relative Humidity (%) |
|---|---|---|---|---|
|  | $\mu \pm \sigma$ (min − max) | $\mu \pm \sigma$ (min − max) | $\mu \pm \sigma$ (min − max) | $\mu \pm \sigma$ (min − max) |
| **Winter** | 0.5 ± 0.3 (0.025 − 1.6) | 3.5 ± 2 (0.25 − 11) | 7 ± 5 (-7.0 − 16) | 27 ± 10 (8.0 − 88) |
| **Spring** | 0.4 ± 0.3 (0.030 − 1.6) | 3.6 ± 2 (0.0 − 18) | 10 ± 6 (-3.7 − 21) | 40 ± 20 (0.0 − 93) |
| **Summer** | 0.4 ± 0.3 (0.030 − 1.5) | 3.0 ± 1 (0.0 − 10) | 23 ± 4 (11 − 29) | 33 ± 20 (0.0 − 86) |
| **Fall** | 0.4 ± 0.3 (0.005 − 1.7) | 3.4 ± 2 (0.0 − 11) | 15 ± 4 (0.5 − 23) | 25 ± 10 (0.0 − 92) |



### 2.3 Eddy Covariance Measurements

Surface-atmosphere exchange was measured using eddy covariance flux techniques. The vertical flux ($F_c$) in this technique is determined by the covariance of the vertical wind speed ($w$) and a scalar ($c$; particle concentration):

$$F_c = \overline{w'c'} = \frac{1}{N} \sum_{i=0}^{N}(c_i - \bar{c})(w_i - \bar{w}) \tag{1}$$

where $N$ is the number of points, $w_i$ and $c_i$ are the instantaneous measurements of the vertical windspeed and particle concentration respectively, and $\bar{w}$ and $\bar{c}$ are the mean values. We used a flux averaging time interval of 30-min.
Deposition velocity can be derived from the vertical flux and the mean concentration over a given period.

$$F_c = -V_{dep}\, \bar{c} \tag{2}$$

Using this notation, a positive $V_{dep}$ corresponds to deposition and a negative $V_{dep}$ indicates an emission. We treat the positive and negative fluxes as two separate processes following uneven distribution of positive and negative fluxes around zero (**Figure S3**). Only negative fluxes were used to derive deposition velocities, consistent with other
particle flux studies (Emerson et al., 2020; Lavi et al., 2013). Pryor et al. (2013) identified drivers of positive fluxes in a pine forest, which gives additional support to the separation of positive and negative fluxes in the calculation of deposition velocities ($V_{dep}$). Additionally, the variability in some of the seasonal measurements was high, resulting in means either at the edge or outside the 25[th] to 75[th] confidence intervals; we thus use the medians of the deposition velocity data to investigate changes between seasons.

### 2.3.1 Data Treatment and Quality Control

Several quality controls were enforced on the data based on $u*$, stationarity, and precipitation events. We also removed data taken during exceptional events. For example, during the fall campaign period, several prescribed burns were carried out in an area adjacent to MEFO. We exclude particle flux data from time periods in which CO
concentration was elevated (> 3800 ppb), which resulted in the removal of data from 10[th] October. Data that did not meet the following requirements for $u*$, stationarity, and precipitation quality control were rejected:

1. Periods in which turbulence was not well developed, defined by a friction velocity ($u*$) < 0.14 m/s, were excluded (Papale et al., 2006; Reichstein et al., 2005).
2. Measurement periods in which the flux was not in steady state, as determined by a stationarity test, were
excluded as they fail the assumptions of the eddy covariance method. A stationarity test compares 5-min fluxes to the full 30-min flux to ensure that the fluxes do not vary during the period of interest (Foken and Wichura, 1996). A period is considered to have stationarity if the 5-min periods do not deviate from the 30-min period by more than 30%, and the following criterion is met:

$$0.7 < \frac{\overline{\langle w'c' \rangle}_{5min}}{\overline{\langle w'c' \rangle}_{30min}} < 1.3 \tag{3}$$

3. Precipitation events were excluded from this analysis, as they can affect the signal of the sonic anemometer and distort the measured flux. A total of 24 precipitation events occurred, 3 in the winter, 12 in the spring, 9 in the summer, and zero in the fall.

These filters resulted in 241, 180, 305, and 363 flux periods for winter, spring, summer, and fall respectively. Results of each test and the number of flux periods impacted is summarized in **Table 2** (**Figure S3**).






**Table 2**: Summary of quality control tests for the SPiFFY campaign. Number of flux periods that did not meet the quality control factor listed as well as the percent of the total available measurements that did not meet the standard, are presented for each test and each season. The original number of measurement periods is listed under the season headers, and the number of flux periods retained are listed in the final row.

|  | Winter (N=858) | Spring (N=1001) | Summer (N=1330) | Fall (N=1433) |
|---|---|---|---|---|
| $u*$ | 76 (9%) | 141 (14%) | 198 (15%) | 231 (16%) |
| Stationarity | 513 (60%) | 722 (72%) | 965 (73%) | 1019 (71%) |
| Precipitation | 96 (11%) | 279 (28%) | 0 (0%) | 12 (0.01%) |
| **Accepted Flux Periods** | **241 (28%)** | **180 (18%)** | **305 (23%)** | **363 (25%)** |


### 2.3.2 Corrections

A single point storage correction was applied to the data in order to account for the difference in turbulent flux bellow the measurement height (Rannik et al., 2009).


$$F_{storage} = \int_0^{z_r} \frac{\delta \bar{c}}{\delta t} \, dz \approx \frac{\overline{c(t+\Delta T)} - \overline{c(t)}}{\Delta T} \qquad (4)$$

This storage correction resulted in a < 1% change in the total flux in all four seasons.

Additionally, a two-dimensional rotation of windspeed in three axes corrected for the sonic anemometer not being mounted with a perfect level over the footprint (Wilczak et al., 2001; Massman, 2000).

### 2.3.3 Signal-to-Noise and Flux Uncertainty


To account for uncertainty in calculated flux measurements, we considered the signal-to-noise ratio of the UHSAS as well as flux uncertainty from instrument noise, counting statistics, and the covariance measurement. The signal-to-noise ratio of the UHSAS number concentration measurements is defined as the ratio between the mean concentration ($\mu$) of a period and the standard deviation of the instrument signal during a system zero ($\sigma_{zero}$). A system zero is defined as a measurement period in which a HEPA filter is placed in front of the inlet. Here an adjacent period to the system zero was used to calculate the signal-to-noise ratio in each season (**Figure S4**).


$$SNR = \frac{\mu}{\sigma_{zero}} \qquad (5)$$

Contribution of instrument noise to the flux uncertainty ($\delta F_{noise}$) was determined using the method described by Billesbach (2011). In this method it is assumed that the contribution of instrument noise to the total uncertainty is the covariance when the correlation coefficient is minimized, which is achieved by randomizing the time sequence of the scalar, or particle concentration.


$$\delta F_{noise} = \frac{1}{M} \sum_{i,j=1}^{M} w'(t_i) \, c'(t_j) \qquad (6)$$

where $M$ is the number of measurements in the interval, $w'$ and $c'$ are the deviations from the mean vertical windspeed and concentration, and $i$ and $j$ are the time indices.


The uncertainty in the flux from counting discrete particles ($\Delta F_N$) is calculated using the cumulative number of particles ($N$), along with the mean concentration ($\bar{c}$), and the variance of the vertical velocity ($\sigma_w$) during a flux period (Fairall, 1984).

$$\Delta F_N = \frac{\sigma_w \bar{c}}{\sqrt{N}} \qquad (7)$$

Finally, the uncertainty in the covariance is quantified here using the method outlined by Finkelstein and Sims (2001). Estimation of the random error comes from the calculation of variance of a covariance when the two






variables, here vertical windspeed and particle concentration, are lagged at unrealistic time scales (50 – 60s). Finkelstein and Sims (2001) outline the following parameterization for the variance of the covariance ($\sigma^2$):

$$\sigma^2 = \frac{1}{M}\left[\sum_{i=-m}^{m}\sigma_{x,x}^2(i)\,\sigma_{y,y}^2(i) + \sum_{i=-m}^{m}\sigma_{x,y}^2(i)\,\sigma_{y,x}^2(i)\right] \tag{8}$$

where $M$ is the number of measurements in a flux period, $\sigma_{x,x}^2$ and $\sigma_{y,y}^2$ are the variance of the two variables, $\sigma_{x,y}^2$ and $\sigma_{y,x}^2$ are the estimated covariances of the two variables, and $m$ is number of samples used to captures the integral time scale ($m = 200$, 20s of 10 Hz data). The auto- ($\sigma_{x,x}^2$) and cross-covariance ($\sigma_{x,y}^2$) is computed for a lag ($h$) by:

$$\sigma_{x,x}^2(h) = \frac{1}{M}\sum_{i=1}^{M-h}(x_t - \bar{x})(x_{t+h} - \bar{x}) \tag{9}$$

$$\sigma_{x,y}^2(h) = \frac{1}{M}\sum_{i=1}^{M-h}(x_t - \bar{x})(y_{t+h} - \bar{y}) \tag{10}$$

The subsequent uncertainty in the covariance ($\sigma_{w'N'}$) ranged from 30 – 80 # cm$^{-2}$ s$^{-1}$. We also evaluated the time-
lagged covariance spectra for each flux period out to 50s to attempt and identify the time lag between the vertical windspeed and particle concentration measurements. However, the determination of a time lag by cross covariance is problematic for data limited by counting statistics (Langford et al., 2015) so a fixed lag time of 3.5 s was used based off the flow through the inlet line. We acknowledge that the use of a fixed lag time can lead to flux underestimation as it does not capture changes in the system over time. For all the calculations outlined above, some
variable notation has been changed from the original source in order to have consistent references in this work. Results of the error analysis are summarized in **Table 3**. The instrumental and random noise were both within the measured variation of the particle fluxes, however, the uncertainty from counting discrete particles exceeded the measured variation. This indicates that the uncertainty from counting is the overwhelming contributor to uncertainty in these flux measurements.

**Table 3:** Summary of uncertainty and LOD for total particle flux measurements during SPiFFY

| | Winter | Spring | Summer | Fall |
|---|---|---|---|---|
| \|Total Flux\| ($\mu \pm \sigma$; # cm$^{-2}$ s$^{-1}$) | ($2 \pm 70$) | ($60 \pm 100$) | ($50 \pm 100$) | ($20 \pm 100$) |
| *SNR* | 100 | 200 | 300 | 300 |
| $\delta F_{noise}$ (# cm$^{-2}$ s$^{-1}$) | 20 | 30 | 30 | 20 |
| $\Delta F_N$ (# cm$^{-2}$ s$^{-1}$) | 800 | 2000 | 2000 | 1000 |
| $\sigma_{w'N'}$ (# cm$^{-2}$ s$^{-1}$) | 30 | 80 | 70 | 60 |
| $\overline{LOD_{season}}$ (# cm$^{-2}$ s$^{-1}$) | 8 | 40 | 20 | 20 |





### 2.3.4 Flux Limit of Detection

Limits of detection for individual flux periods ($LOD_i$) were first calculated considering random error using the method from Langford et al. (2015):

$$LOD_i = \alpha\, RE_i \tag{11}$$

with $\alpha$ being the specified confidence interval ($\alpha = 3$ was used for the 99th percentile in this work), and $RE_i$ representing the random error of the flux period from the Finkelstein and Sims (2001) calculation above ($\sigma_{w'N'}$).

Time-resolved limits of detection were used to calculate an average limit of detection for each season, $LOD_{season}$.

$$\overline{LOD_{season}} = \frac{1}{N}\ \sqrt{\sum_{i=1}^{N} LOD_i^2} \tag{12}$$

For this project, the average limits of detection for particle flux measurements using this method were 8 (winter), 40 (spring), 20 (summer), and 20 (fall) # cm$^{-2}$ s$^{-1}$. No flux periods were excluded based on comparison to $\overline{LOD}_{season}$ or time dependent $LOD_i$ values as they still provide useful information when averaged (Langford et al., 2015). These

numbers are high compared to an $LOD$ calculated through flux analysis on zero periods, which resulted in $\overline{LOD}_{zero}$ of 4 # cm$^{-2}$ s$^{-1}$ for both the winter and fall. While the $\overline{LOD}_{zero}$ needs further investigation if it is to be verified because it requires the attachment of a HEPA filter to the front of the inlet, which could change the turbulent dampening through the inlet and therefore require spectral correction before use. However, the large difference between $\overline{LOD}_{season}$ and $\overline{LOD}_{zero}$ indicates a need for the critical evaluation of $LOD$ calculation methods for flux

measurements in the future.

### 2.3.5 Spectral Analysis

The UHSAS and other similar optical particle instruments have been previously used for eddy-covariance measurements, and their measurements have been validated using spectral analysis (Petroff et al., 2018; Deventer et

al., 2015). Here, we use spectral analysis to investigate and validate the measurements. Frequency weighted dimensionless cospectra of vertical wind speed and particle concentration followed the sensible heat, and the inertial subrange (f $^{-4/3}$) predicted by Kolmogorov theory is observed for each season between 0.01 and 5 Hz (**Figure 2**). **Figure 2** provides example cospectra for a 30-minute period from each season during the day where $u^* \geq 1$ m s$^{-1}$ to reduce noise caused by low turbulence. Stability during the periods were -0.03, -0.04, -0.10, and -0.04 for the winter,

spring, summer, and fall.



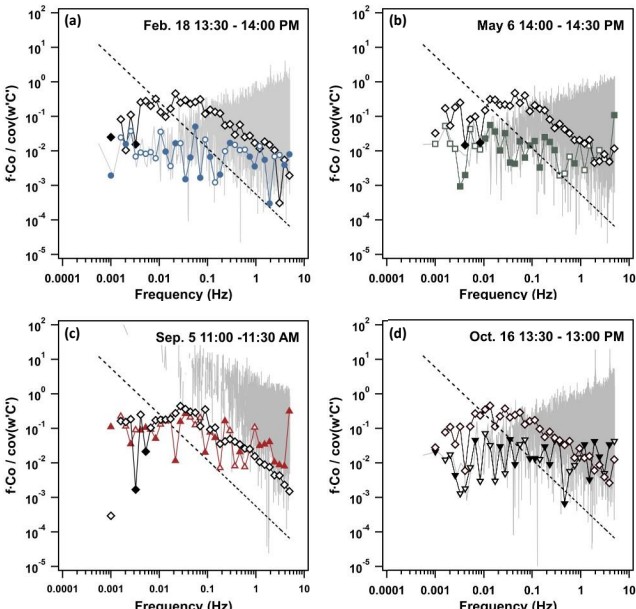

**Figure 2**: Frequency weighted dimensionless cospectral density of particle concentration (wC, color) and temperature (wT, black diamonds) with vertical wind speed for a representative 30-minute period for each of the four seasons. Individual data points are medians from 40 evenly spaced logarithmic bins, with open points representing positive data and closed points representing negative data that have been forced positive. The cospectra are presented for the (a) winter (blue circles), (b) spring (green squares), (c) summer (red upward triangles), and (d) fall (dark brown downward triangles). Raw wC cospectra are shown in light grey in the background and the inertial subrange ($f^{-4/3}$) is shown with the dashed line.

### 2.4 Deposition and Leaf Level Modeling

#### 2.4.1 Deposition Models

We used the single layer resistance model from Emerson et al. (2020), which is based on the models presented by Zhang et al. (2001) and Slinn (1982), and aspects of the multi-layer model proposed by Katul et al. (2010) to investigate the roles of different mechanisms in controlling measured deposition velocities. The full parameterization of Emerson et al. (2020) along with the relevant parameterizations from Katul et al. (2010) are in **Appendix A**. We integrated the work of Katul et al. (2010) into the deposition velocity framework of Emerson et al. (2020) in order to investigate the impact of turbophoresis on particle deposition. Following Zhang and Slinn, the Emerson et al. (2020) framework is based on the gravitational settling velocity ($V_g$), the aerodynamic resistance ($R_a$), and the surface resistance ($R_s$).

$$V_d(d_p) = V_g(d_p) + \frac{1}{R_a + R_s} \tag{13}$$

We then incorporated the term developed by Katul et al. (2010) describing the collection efficiency of turbophoresis ($E_{turbo}$) into the surface resistance parameterization ($R_s$) from the Emerson et al. (2020) model by adding it in series to the collection efficiencies of Brownian diffusion processes ($E_b$), impaction ($E_{im}$), and interception ($E_{in}$) (Eq. 14).

$$R_s = \frac{1}{\varepsilon_0 \, u^* \, (E_b + E_{im} + E_{in} + \boldsymbol{E_{turbo}}) \, R_1} \tag{14}$$



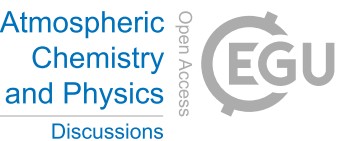

$$E_{turbo} = \frac{\tau_p}{1 + \frac{\tau_p}{\tau}} \left( \frac{\sigma_w^2}{\pi \, b \, \eta} \right)$$    (15)

The new surface resistance relies on the previously listed collection efficiencies, $u^*$, and the bounce correction ($R_1$) along with the empirical constant $\varepsilon_0 = 3$. $E_{turbo}$ is dependent on the particle relaxation time ($\tau_p$), Lagrangian turbulent timescale ($\tau$), the standard deviation of the vertical velocity ($\sigma_w$), the thickness of the viscous sublayer for the vertical velocity variance ($b$), which can range from $5 < b < 50$, and the kinematic viscosity of air ($\eta$). In the

parameterization for $E_{turbo}$, the particle relaxation time ($\tau_p$) is the size dependent term (**Appendix A**). While we were able to use measured values for $\sigma_w$ and in the calculation of the Lagrangian turbulent timescale, these values can be approximated using $u^*$. Standard deviation of the vertical velocity trends linearly with $u^*$, with an acceptable approximation being $\sigma_w \approx 1.0 \, u^* - 1.2 \, u^*$ (Finnigan, 2000). Data from all four measurement periods of the SPiFFY campaign had a relationship of $\sigma_w \approx 1.1 \, u^*$ (**Figure S5**). Additionally, the Lagrangian turbulent timescale

can be approximated by $\tau \approx (0.3 \, u^*) / (1.1 \, u^*)^2$. This approximation relies on the $\sigma_w / u^*$ relationship from before as well as $u^* / \overline{U} \approx 0.3$ (Poggi et al., 2004; Finnigan, 2000), which we were also able to validate using the SPiFFY data (**Figure S6**).

### 2.4.2    Leaf Level Energy Balance and Thermophoretic Settling Velocity

We used a simple leaf energy balance to explore leaf effects on particle deposition. The energy balance calculations were based on equations outlined by Monteith (1990), Sridhar and Elliott (2002), and Jones (2014). We used the open-source single point leaf energy balance framework developed by Kevin Tu (http://landflux.org/Tools.php) to help structure the energy balance. We adapted these equations and frameworks to work with real time meteorology data reported by the U.S. Forest Service (Frank et al., 2021) along with sonic anemometer data collected during the

campaign periods. A full description of the leaf energy balance calculation is in **Appendix B**.

Thermophoretic settling velocity was calculated following Salthammer et al. (2011) and Hinds (1999). This velocity was then integrated into the Emerson et al. (2020) deposition model (**Appendix B**).

### 3    Results and Discussion

### 3.1    Seasonal trends in particle fluxes and concentrations


Total (0.08–1 µm) particle concentration (# cm$^{-3}$), flux (# cm$^{-2}$ s$^{-1}$), and exchange velocity ($V_{ex}$, cm s$^{-1}$) all exhibited distinct diel cycles (**Figure 3, Figure S7**). The flux and $V_{ex}$ magnitude peak in the middle of the day due to increased turbulence (see $u^*$ in **Figure 1**), while particle number concentration peaks at night and decreases during the day as the boundary layer grows and mixes with larger volumes. Daytime fluxes were largest in the summer and spring

measurement periods, with nighttime fluxes being comparable across all four seasons. Flux measurements were dominated by downward fluxes in all seasons, except the winter, resulting in negatively skewed flux data (**Figure S3**).

As discussed in the introduction, particle concentration and size distribution can impact the direction and magnitude of particle fluxes. Particle concentrations were highest in the summer and lowest in the winter, providing a partial

explanation for the larger particle fluxes in summer versus winter. As exchange and deposition velocities are independent of concentration, seasonal shifts in size distribution could potentially account for the observed seasonality in these values. However, no major changes in the average size distribution occurred between seasons (**Figure 3**). Daytime count median diameters were $128 \pm 8$, $140 \pm 20$, $140 \pm 10$, and $130 \pm 10$ nm for the winter, spring, summer, and fall, respectively. These count median diameters did not change substantially at night (**Figure**

**S8**, **Table S1**). Particle distributions did not change between periods of positive and negative flux (**Figure S9**, **Table S2**). This consistency in the distributions indicates that changes in total particle exchange velocities are not attributable to seasonal shifts in size distribution.

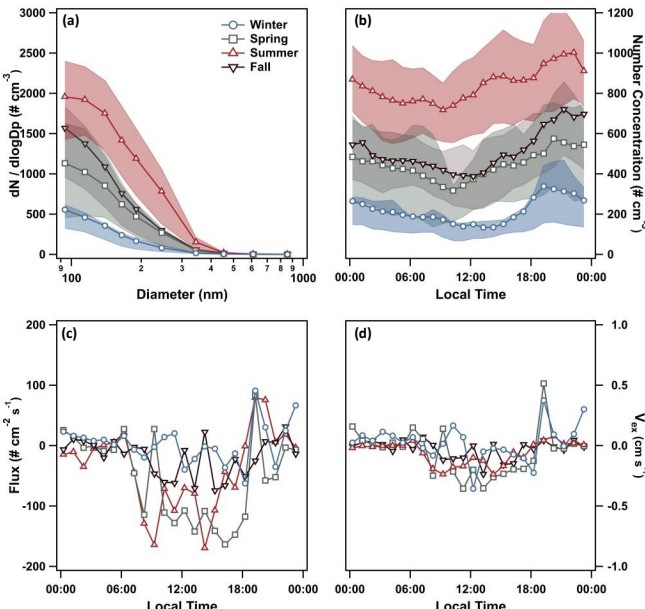

**Figure 3**: (a) Average daytime size distribution (dN/dlogDp) of particles for each season, with the 25[th] to 75[th] confidence intervals shaded, as well as diel trends of hourly mean (b) total number concentration (also with the 25[th] to 75[th] confidence intervals shaded), (c) flux, and (d) exchange velocity ($V_{ex}$) across seasons.

### 3.2 Measured and modeled seasonal trends in particle dry deposition

As noted above, we segregate the downward flux periods from periods in which upward fluxes occurred and used these downward fluxes and their associate particle number concentrations to calculate deposition velocity, $V_{dep}$. Our sign convention now switches to give positive velocities for $V_{dep}$. Deposition velocity for all particles measured followed the diel trend of both friction velocity and sensible heat flux, with peak values occurring around noon (**Figure 4**, **Figure S10**). Additionally, size-dependent deposition velocities in each season followed trends previously observed in needleleaf forests (Lavi et al., 2013; Mammarella et al., 2011; Vong et al., 2010; Grönholm et al., 2009a; GröNholm et al., 2007; Gallagher et al., 1997; Lorenz and Murphy, 1989). Binned deposition velocity from all seasons had a strong linear relationship with friction velocity (**Figure 5**); this relationship was present in all size ranges measured during the study with larger particles having bigger slopes (**Figure S11, Table S3**). This trend is consistent with other particle deposition studies (Petroff et al., 2018; GröNholm et al., 2007).

While total deposition velocity did not vary significantly between seasons, the size-dependent deposition velocity in the winter was greater than in the spring, summer, and fall. The enhancement of winter deposition relative to summer was the largest and ranged from 23 – 202% (130 ± 60%) depending on particle size, with particles less than 0.35 µm having the largest enhancement. This enhancement decreased when deposition was normalized by $u*$, ranging from 17 to 145 % (80 ± 40%), but was still significant based on t-tests of the size-dependent data (only the 0.505 – 0.711 µm range did not show significant difference). Wintertime enhancement of particle dry deposition velocities has been observed in two other particle deposition studies. First, Suni et al. (2003) compiled six years (1996 - 2001) of particle flux measurements (total particle number from 0.014 - 3 µm) over a Scots pine forest in Hyytiälä, Finland, from 1996 to 2001. They hypothesized that the larger wintertime bulk deposition was a result of a larger presence of small particles (< 0.10 µm) having larger deposition velocities, that when included in the integrated total deposition velocity increased wintertime measurement. Second, Rannik et al. (2009) presented particle flux measurements from the same site as Suni et al. (2003) but for 2000 to 2007. In contrast to Suni et al.,

Rannik et al. (2009) noted the same increase in wintertime deposition, but found that the number of nucleation days – on which small particles would dominate – was lowest in the winter. Additionally the continued analysis by Mammarella et al. (2011), indicated that the seasonal changes in geometric mean diameter were inadequate to cause the increased winter deposition rates. Similar to our observations, Mammarella et al. (2011) found that size-dependent deposition velocities were significantly higher in winter than other seasons. That work concluded that the observed seasonal dependence of the dry deposition was driven by bi-modal distributions in the wintertime.

We investigated the cause of the wintertime increase in deposition during SPiFFY by comparing our results to the resistance model from Emerson et al. (2020), which is based on the work of Zhang et al. (2001) and Slinn (1982). This model framework was chosen as the base comparison because of its wide use in chemical transport and climate models, including GLOMAP and GEOS-Chem, as well as its ability to assess the roles of Brownian motion, gravitational settling, interception, and impaction. Using measured values $u*$, windspeed, temperature, and stability function, we evaluated the ability of the Emerson et al. (2020) model to capture seasonal variation in both total and size-dependent dry deposition (**Figure 6**). While the model accurately captured the diel trends for the summer data, there was a clear systematic underestimation of deposition in the other seasons. The largest underestimation was in the winter, when the predicted deposition was ~90% lower than the measured values (**Figure S12**). The size-resolved model predictions clearly disagreed with the measured winter deposition in every size bin. To resolve this disagreement we investigated additional mechanisms that are not currently accounted for in the Emerson et al. (2020) parameterization. Additionally, we explored possible seasonal dependencies for terms that are currently considered, but not parameterized to be seasonally dependent (**Section 3.3**).

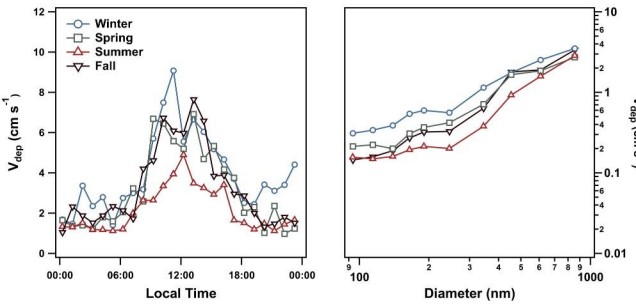

**Figure 4**: Diel cycle of median total particle dry deposition (left), and median size-dependent dry deposition for each season (right).

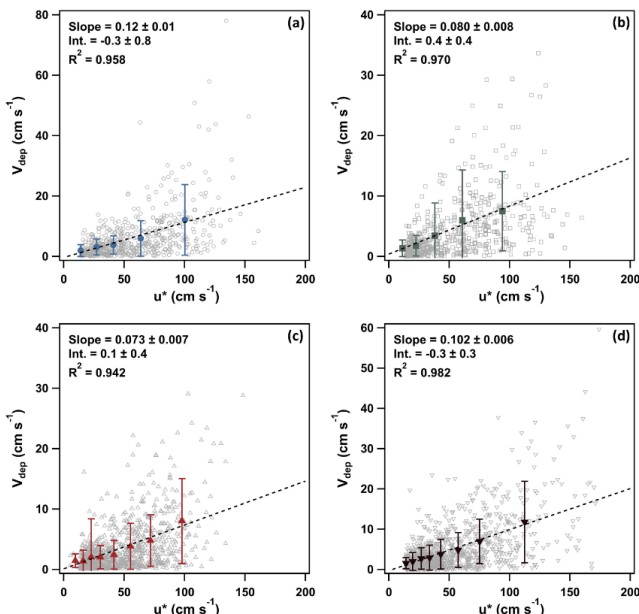

**Figure 5**: Total deposition velocity verses friction velocity ($u*$) for the (a) winter, (b) spring, (c) summer, and (d) fall. Grey points are raw data, and the colored markers are mean deposition binned by $u*$ (each one representing 200 measurement points), with the error bars representing standard deviation.

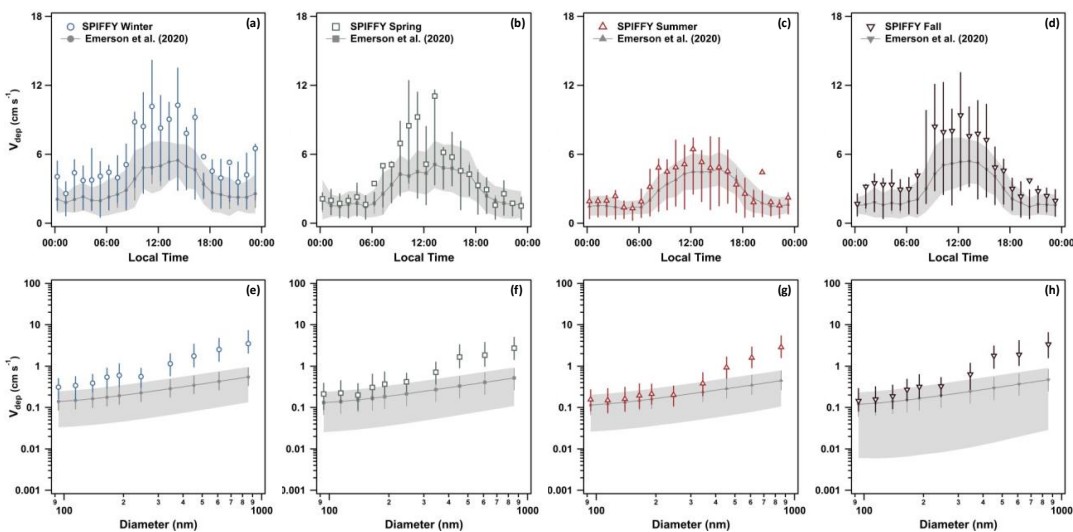

**Figure 6**: Average measured diel deposition velocity compared to the modeled diel deposition for the (a) winter, (b) spring, (c) summer, and (d) fall using the resistance model from Emerson et al. (2020). Additionally, seasonal size-dependent deposition trends compared to the model are shown for all seasons in panels (e) through (h). For all plots the modeled deposition is shown as closed grey markers, with shading representing the standard deviation and grey bars representing the interquartile range. Measured average data are represented as open markers with bars representing the interquartile range.



### 3.3 Influence of scalar gradients on seasonal deposition velocities

Phoretic effects, which are the drift of particles induced by gradients in scalars such as temperature, water vapor, and electricity, can impact particle movement. The influence of these gradients has been well studied in indoor environments (Salthammer et al., 2011), however, their contribution in outdoor forested environments is still uncertain (Farmer et al., 2021; Petroff et al., 2008). Previous studies have hypothesized that such gradients could impact particle deposition over snow and ice surfaces and impact forested environments during the winter. In their deposition model, Petroff and Zhang (2010) used a constant parameter to describe bulk phoretic effects over ice, snow, and water surfaces; this single phoretic parameter helped resolve differences between measured and modeled deposition velocities. Mammarella et al. (2011) used their measurements of particles in the Brownian diffusion dominated regime ($0.020 - 0.065$ µm) to investigate the power law dependency between normalized deposition velocity ($V_{dep}/u^*$) and the Schmidt number ($Sc$) and found that this relationship differed for winter versus other seasons. While Mammarella et al. (2011) ultimately concluded that the strong winter deposition velocities must be due seasonal differences in size distribution, they noted that additional factors, such as electrostatic and thermophoretic forces, could contribute to the enhancement of deposition of very small particles. Here, we hypothesized that snow covered canopies during SPiFFY were affected by phoretic effects – in particular, gradients in temperature (Section 3.3.2) or turbulence (Section 3.3.2). We do not consider electrophoresis, as the findings of Tammet el al. (2001) indicate that these gradients only affect $0.01 - 0.2$ µm particles above the canopy during low-wind conditions.

### 3.3.1 Thermophoretic effects on deposition

We first investigated thermophoresis, which is the drift of particles caused by temperature gradients between the air and collecting surfaces, as a possible driver of the increased winter deposition caused by the low temperatures and the addition of snow to the canopy (Batchelor and Shen, 1985). We modeled these gradients using measured meteorological parameters along with a simple leaf level energy balance (**Appendix B**). This energy balance showed that in the winter, needles were consistently colder than the surrounding air during the day, while temperatures at night were comparable. Specifically, the leaf level energy balance predicted an average gradient of 4 ± 2 K between the leaf and surrounding air. Incorporating the thermophoretic settling velocity with this gradient into the Emerson et al. (2020) deposition model yielded negligible changes in the predicted particle deposition velocity. In the sub-micron size range of interest here, thermophoresis is not strongly correlated to particle size. Unrealistic gradients ($15 - 60$ K/mm) would be needed for thermophoresis to drive the observed change in size-dependent deposition. Thus, thermophoresis cannot compete with the other drivers of deposition in a needleleaf forest.

### 3.3.2 Movement of particles by turbophoresis

Discrepancies between models and field observations of particle deposition are common, and often attributed to missing deposition mechanisms (Saylor et al., 2019; Pryor et al., 2008). Movement of particles from areas of high to low turbulence, or turbophoresis, has been proposed as an important mechanism for dry deposition of particles in the accumulation mode (Mammarella et al., 2011; Katul and Poggi, 2010; Katul et al., 2010; Feng, 2008). Two studies successfully integrated a turbophoresis term into a size-resolved dry deposition parameterization. First, Feng (2008) developed a zero-dimensional representation of particle deposition based on the deposition in pipes. In that model, turbophoresis – referred to as the "burst effect of atmospheric eddy turbulence" – is parameterized according to the roughness Reynolds number. Later, Katul et al. (2010) developed a multi-layer model for particle dry deposition. In the Katul model, turbophoresis was parameterized according to the vertical momentum flux, or turbulent stress, and the standard deviation of the vertical velocity. Katul et al. (2010) concluded that the effects of turbophoresis were most prominent for particles between 0.1 and 10 µm above and in the upper layers of the canopy, and that the effects could be neglected for particles < 0.01 µm.

To investigate the effect of turbophoresis on the SPiFFY observations, we used the Katul et al. (2010) parameterization (**Appendix A**) because it maintains the minimum in size-dependent deposition velocity, in contrast to the Feng (2008) model. This minimum is a key characteristic of particle deposition that has emerged in recent syntheses of observations (Farmer et al., 2021; Saylor et al., 2019; Hicks et al., 2016). However, the Katul





parameterization has two components that may result in underestimation of deposition to the canopy. First, the model neglects interception, which has a significant role in deposition over forests. Second, the model relies on the inertial impaction term from Slinn and Slinn (1980), which was formulated for water and smooth surfaces. Katul et al. (2010) acknowledged these as possible reasons for underestimation of particle deposition and recognized the need to further consider "microroughness" of leaves and needles.

Single point calculations of turbophoretic velocity ($V_t$), following those of Katul et al. (2010), allow us to bound the possible contribution to total deposition. Turbophoretic velocity ($V_t$) for a 0.1 µm particle ($\tau_p = 1.4 \times 10^{-7}$ at 298 K) should be ~0.04 cm s$^{-1}$ assuming a u* = 1.0 m s$^{-1}$ and a leaf level boundary layer thickness of 0.15 mm. This is an order of magnitude lower than our measured deposition velocities for this size range and is therefore expected to be negligible. Using the same assumptions but for a 1 µm particle ($\tau_p = 5.0 \times 10^{-6}$ at 298 K) we estimate that $V_t$ is ~1.3

cm s$^{-1}$, potentially resulting in a ~40% increase in the total modeled deposition for particles in that size range. This single point calculation based on the Katul parameterization indicates that turbophoresis may play a role in deposition of larger particles ($\geq 1$ µm). However, turbophoresis is unlikely to fully explain the seasonal differences observed during SPiFFY as the largest seasonal discrepancies occurred at the lower end of our size range.

      To investigate the impact of turbophoresis over the entire size range and over time while accounting for challenges

with the Katul et al. (2010) model, we isolated the turbophoretic collection efficiency from the model and integrated it into the Emerson et al. (2020) parameterization of surface resistance (**Section 2.4.1**). We present both the resulting deposition velocities ($V_{dep}$) and surface resistances ($R_s$) – presented as velocities – from the model with ($V_{dep} + E_{turbo}$, $R_s + E_{turbo}$) and without ($V_{dep}$, $R_s$) turbophoresis in **Figure 7a**. As a sensitivity test, we varied the thickness of the viscous sublayer (*b*) between 5 and 50 mm following Katul et al. (2010). Turbophoresis resulted in

large changes for deposition of 1 – 10 µm particles, with larger changes for a thinner viscous sublayer, i.e. when *b* = 5 (**Figure 7a, 7b**). The incorporation of turbophoresis resolved some of the initial model-measurement disagreement shown in in Figure 6 for larger particles but had little effect on the model-measurement disagreement at the lower end of the measured size range. Even with the addition of turbophoresis, total deposition in the wintertime was still underpredicted by ~40% (**Figure 7c, Figure S13**). Interestingly, inclusion of the turbophoresis term also created a

plateau in deposition for particles greater than 10 µm (**Figure 8**). This modeled plateau is consistent with a feature frequently noted in size-resolved deposition velocity observations, particularly over needleleaf forests. Saylor et al. (2019) developed an empirical logistic equation to resolve this feature but attributed it to an unknown mechanism. Turbophoresis thus provides a potential mechanism to explain this observed feature in 1 – 10 µm size range. A more critical exploration of the turbophoresis parameterization would be needed to resolve this.

Both the Emerson et al. (2020) model and the integrated Emerson et al. (2020) with $E_{turbo}$ were compared against literature values for size resolved deposition in forests (Zhang et al., 2014; Lavi et al., 2013; Gordon et al., 2011; Mammarella et al., 2011; Vong et al., 2010; Grönholm et al., 2009b; Pryor et al., 2007, 2009; GröNholm et al., 2007; Pryor, 2006; Gaman et al., 2004; Gallagher et al., 1997; Lorenz and Murphy, 1989; Waraghai and Gravenhorst, 1989; Grosch and Schmitt, 1988; Höfken and Gravenhorst, 1982). The two parameterizations were

both comparable to the synthesized literature data, which can be observed in **Figure 8**. For these comparisons the models were run for both a needleleaf and broadleaf condition in the midsummer, assuming u* = 0.5 m s$^{-1}$, T = 20 °C, and an average wind speed of 3 m s$^{-1}$.

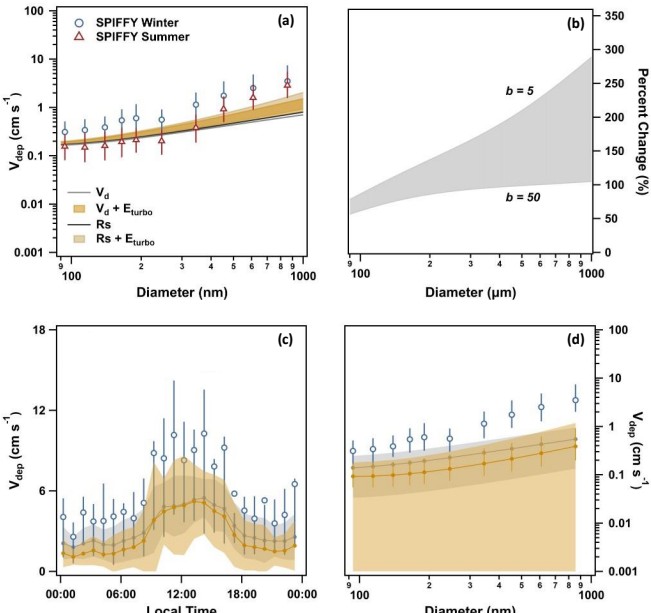

**Figure 7**: (a) Size resolved deposition measured during the winter (blue circles) and summer (red triangle) periods
of SPiFFY compared to modeled deposition velocities from Emerson et al. (2020) ($V_d$; light grey line) versus
Emerson et al. (2020) with the inclusion of turbophoresis collection efficiency in the surface resistance term ($V_d$ +
$E_{turbo}$; gold shaded region). Surface resistance without turbophoresis ($R_s$; dark grey line) and with turbophoresis
incorporated ($R_s + E_{turbo}$; dark gold shaded region) are shown as velocities. The shaded range shows how the second
model formulation changes with the thickness of the viscous sublayer ($b$); higher deposition values occur at smaller
$b$ values. (b) The percent increase in predicted deposition with the addition of turbophoresis is shown as a function
of particle size. (c) The average observed (blue circles) versus predicted (shaded range) diel cycle and (d) size-
dependent prediction of the model with (grey circles, grey shading) and without (gold circles, gold shading)
turbophoresis considered is shown for the winter. The shading represents standard deviation; bars on data points
represent the interquartile range.

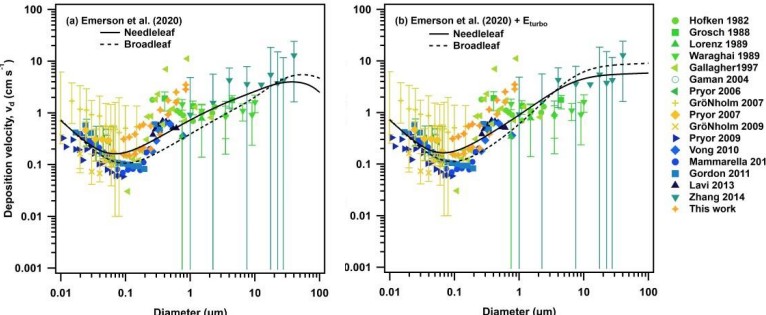

**Figure 8**: Comparison of  (a) the Emerson et al. (2020) model alone and (b) the Emerson et al. (2020) model with
the turbophoresis collection efficiency ($E_{turbo}$) from Katul et al. (2010) integrated into the surface resistance ($R_s$)
term.



### 3.4    Influence of snow cover on deposition velocity


Thermophoresis and turbophoresis are not the only hypotheses proposed in the literature to explain seasonal variation in particle deposition velocities. Gallagher et al. (1992) observed a significant decrease in $3 - 31$ µm cloud droplet deposition over a snow-covered Sitka spruce canopy. That study used an isolated snow event to observe the effect of snow on deposition and found deposition was two times lower during snow cover. This snow-driven

suppression in $V_{dep}$ was attributed to a decrease in surface roughness of the canopy and the subsequent increase of the effective target diameter. Changes in surface roughness from snowfall are not directly accounted for in either the Emerson et al. (2020) or the Zhang et al. (2001) model, and could contribute to model-measurement disagreement. In both model parameterizations, roughness length is used to describe surface roughness and is defined with a seasonally variable term. For an evergreen needleleaf forest, the roughness length ranges from $0.8 - 0.9$ m, with a

drop in the term during the midsummer and transitional spring, but not during the winter period. This approach is consistent with our measured roughness length, which did not vary substantially between seasons. During the spring SPiFFY campaign, we captured an isolated snow period, allowing us to contrast snow-covered fluxes versus prior bare forest surfaces. We observe no significant differences in the total or size-dependent deposition trends during these two periods (**Figure 9**), indicating that another mechanism, independent of snow, is driving the increased

wintertime deposition velocities.

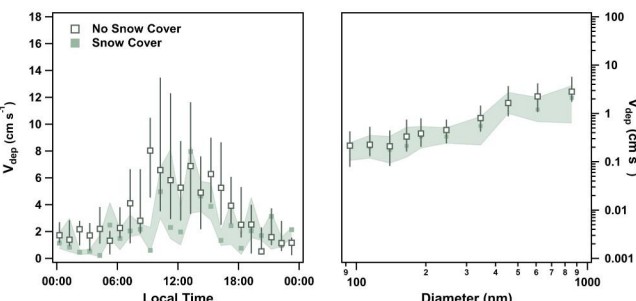

**Figure 9**: Average diel cycle of total deposition velocity (left) and average size-dependent deposition velocity (right) during periods of snow cover (solid green markers with shaded interquartile range) versus no snow cover (open markers with bars representing the interquartile range) from the spring.

### 3.5    The role of interception and changes in the needle surface during the winter

While many deposition studies have proposed alternate mechanisms to explain the numerous model-measurement discrepancies, few have suggested tuning currently considered mechanisms by season. Below we explore aspects of dry deposition mechanisms that could be seasonally dependent. We investigate the extent to which seasonality in

certain deposition mechanisms and their associated parameterizations could explain the enhanced wintertime deposition velocities observed at Manitou. We do not consider roughness length ($z_0$) nor the characteristic radius of the collectors ($A$); while both terms are well established to be seasonally dependent over deciduous forests, measured roughness length did not vary strongly between seasons at SPiFFY (**Figure S2**) and the characteristic radius for an evergreen needleleaf forest is not expected to change seasonally. We first consider potential seasonality

in Brownian diffusion, and then interception.

As discussed above, Mammarella et al. (2011) indicated that deposition of smaller particles ($0.020 - 0.065$ µm) behaved differently in winter than in other seasons. Because the deposition of these particles is dominated by Brownian diffusion, the group investigated potential seasonality in that term. Following previous work, Mammarella et al. parameterized the Brownian collection efficiency as:

$$E_b = C_b Sc^{-\gamma} \qquad (16)$$



where $C_b$ and $\gamma$ are constants. Proposed values for $\gamma$ have historically been land use dependent but not seasonally dependent. For example, Slinn and Slinn (1980) recommend a $\gamma = 1/2$ for water surfaces and Slinn (1982) proposed that $\gamma = 2/3$ for vegetated surfaces. In the Zhang et al. (2001) and subsequent Emerson et al. (2020) model, $\gamma$ is also defined as a land use dependent constant ranging from 1/2 to 2/3. Mammarella et al. (2011) found that scaling
observed deposition velocities with the Schmidt number ($Sc$) implied a much lower exponential term of $\gamma = 0.36$ in the winter, in contrast to $\gamma = 0.66$ in other seasons. While the study theorized that this was due to phoretic effects, our analyses above shows that thermo-, electro-, and turbophoretic effects are unlikely to influence particle deposition in this ultrafine size range ($< 0.1$ µm) under reasonable wintertime conditions in a similarly structured forest. Thus, while seasonally scaling the constants in Brownian collection efficiency terms decreased model-
measurement discrepancy in one instance, there is no mechanistic basis for such a shift – particularly one that only impacts Brownian diffusion related mechanisms. We note that the Schmidt number is the ratio of kinematic viscosity to diffusion coefficients, which do depend on temperature and therefore will change seasonally. However, while the Schmidt number changes by $40 \pm 7\%$ from the coldest to warmest conditions at Manitou this only results in an average change of $2 \pm 2\%$ in the deposition velocities for our measured size range (**Figure S14**). Further, as
our measured particle size range during SPiFFY was larger (0.08 – 1.0 um) than the Mammarella study, merely changing the seasonality of the Brownian diffusion term did not fully account for the observed changes in wintertime data. In fact applying this scaling to the Emerson et al. (2020) model only caused new problems in model disagreement for the SPiFFY data (**Figure S15**).

The seasonal dependence of interception is typically limited in models to considerations of roughness length and
collector radius – but as described above, neither vary substantially enough at Manitou to account for the enhanced wintertime deposition velocities. Plant physiological literature suggests that other aspects of needleleaf forests may undergo seasonal changes that warrant consideration in deposition. Needleleaf plants are efficient at capturing particles (Beckett et al., 1998, 2000) due to their large available surface area, thick and highly structured epicuticular wax, and high stomatal density (He et al., 2020; Räsänen et al., 2013; Sæbø et al., 2012). While surface area of
evergreen needleleaf trees is considered independent of season, stomatal conditions are well-established to change with environmental conditions (Räsänen et al., 2012, 2013). The seasonality and link to particle uptake of the epicuticular wax properties of evergreen needleleaf trees is less well understood, however seasonal changes in needleleaf wax structure have been observed (Altieri et al., 1994) and it has been shown that elevated temperatures can lead to smother wax structures on needles (Apple et al., 2000) Additionally, the wax can be altered and damaged
by uptake of pollutants which has been linked to changes in the plant's particle uptake capacity (Burkhardt and Pariyar, 2014). Studies of particle loadings collected on evergreen needleleaf trees provide some evidence of seasonality in particle collection efficiency (He et al., 2020; Zha et al., 2018; Zhang et al., 2017; Freer-Smith et al., 2005), but many of these studies lack clear linkages to leaf properties or adequate air concentration data to discern enhanced wintertime deposition velocities, rather than just deposition flux. However, changes in stomatal
conditions, epicuticular wax properties, and needle anatomy could be changing the flow of particles around needle surfaces and the collection efficiency of interception. Seasonal changes in plant physiology would trend with interception over other terms because interception is the only mechanism that depends strongly on particle interactions with the collecting surface. In contrast, mechanisms like Brownian motion and impaction both depend on the energy of the particle.

To determine whether seasonal changes in interception collection efficiency could account for enhanced wintertime deposition velocity, we used the Emerson et al. (2020) parameterization first excluding turbophoresis. Doubling the interception constant ($C_{in}$) from 2.5 to 5 closed the gap between the measured and modeled total and size-resolved deposition (**Figure 10, Figure S13**). This increase in the constant indicates that the collection efficiency of interception approximately doubling in the winter. Incorporating both turbophoretic processes and this seasonally
adjusted interception constant produces comparable model-measurement agreement as the isolated adjusted interception constant (**Figure 10, Figure S13**). However, the isolated increased interception constant produced better size-dependent agreement. The hypothesis that seasonally driven changes in plant physiology enhances particle uptake via interception mechanisms during the winter is consistent with SPiFFY data and can be accounted for with seasonally dependent interception terms.

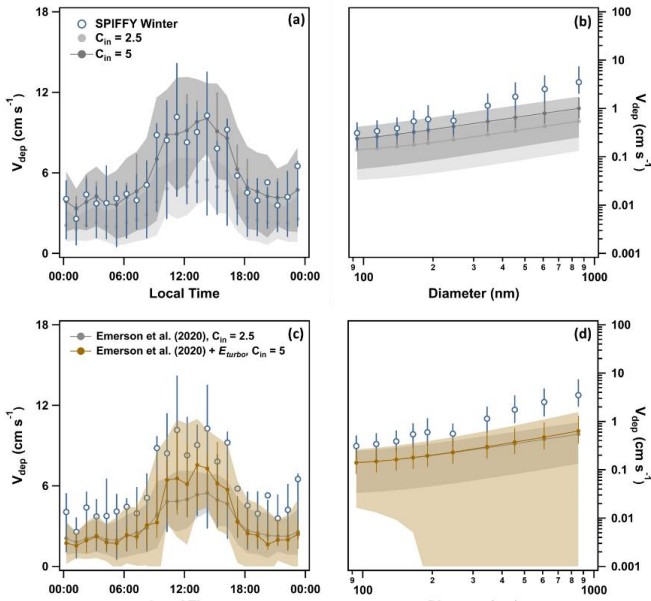


**Figure 10**: Average measured diel deposition velocity (a, c) and size-dependent deposition trends (b, d) compared to the original and adjusted modeled deposition for the winter. The results of the adjusted interception term are shown in (a) and (b), while the results of the adjusted interception term coupled with the inclusion of the turbophoresis term are shown in (c) and (d). For all plots the original modeled deposition is shown as closed light grey markers, with shading representing the standard deviation. The adjusted model results are shown in dark grey (a, b) and yellow (c, d) where average velocities are solid markers, shading represents the standard deviation, and bars represent the interquartile range. Measured average data are represented as open markers with bars representing the interquartile range.

## 4   Conclusions


Significant seasonal changes in particle trends were observed throughout the SPiFFY campaign. Particle concentration, flux, exchange velocity ($V_{ex}$), and deposition velocity ($V_{dep}$) all exhibited similar diel trends that varied in magnitude between seasons. The winter had the lowest concentrations and fluxes of particles, yet the highest deposition velocities; the summer had the highest concentrations and fluxes, with the lowest deposition velocities. The spring and fall acted as transition seasons and fell between the summer and winter trends. These differences extended to the size resolved trends in $V_{dep}$, with the smaller particles (0.1 – 0.3 µm) having the largest changes between seasons. The commonly-used resistance model approach for particle deposition (e.g. Emerson et al. (2020) revision of Zhang et al. (2001)) accurately described size-resolved $V_{dep}$ and total particle $V_{dep}$ in the summer, but was unable to predict seasonal variation or capture $V_{dep}$ in the winter. Trends for total and size-resolved deposition velocity in the winter, spring, and fall were significantly underestimated by the model. The winter had the largest model disagreement (~90%).

The observation of enhanced deposition velocities during winter is consistent with previous particle flux data collected over forests (Mammarella et al., 2011; Rannik et al., 2009; Suni et al., 2003). The wintertime enhancement has been attributed to multiple factors, including increases in concentration of larger particles, scalar gradients (thermophoretic and turbophoretic effects) that drive particles towards surfaces, and changes in surface roughness from snow on the canopy (Mammarella et al., 2011; Rannik et al., 2009; Gallagher et al., 1992). Changes in leaf physiological properties may also enhance particle uptake. However, our quantitative understanding of the relative





impact of these factors is limited (Farmer et al., 2021; Saylor et al., 2019; Hicks et al., 2016). This data set provided
the unique opportunity to probe the importance of these and other mechanisms in controlling deposition velocity
over an evergreen needleleaf canopy across seasons.

The particle size distribution was consistent across seasons and could not explain enhanced wintertime total particle
deposition velocity. Seasonal differences in size-resolved $V_{dep}$ further indicated a missing size-dependent deposition
mechanism. Using a simple leaf-level energy balance and equations outlined by Salthammer et al. (2011) and Hinds
(1999), we ruled out thermophoresis as the source of enhanced $V_{dep}$. We ignored electrophoresis, which has been
previously shown to have a negligible influence on deposition to a canopy (Tammet et al., 2001). Using an isolated
snow event during the spring measurement period, we determined snow in the canopy does not change the surface
roughness enough to impact deposition mechanisms or account for the observed enhancement in wintertime
deposition.

Our analyses show that both turbophoresis and interception can influence dry deposition. Incorporating the
turbophoresis collection efficiency term developed by Katul et al. (2010) into Emerson et al. (2020)'s surface
resistance term improved model-measurement comparisons. Turbophoresis has a greater impact on larger particles
(closer to 1 μm) than smaller, so is not responsible for the enhanced wintertime deposition velocity. The more
interesting result from the incorporation of turbophoresis was the change in shape it created in the predicted size-
dependent deposition trend for large particles (10 – 100 μm). The addition created a plateau in this size range, which
could help explain the shelf observed in deposition measurements over forests for particles > 10 μm. This range was
outside our measured size range, but our results indicate that turbophoresis should be explored further as a
mechanism for dry deposition of particles. We hypothesize that interception in the winter is also enhanced by
changes in the stomatal conditions and needle structure during the winter. Needle surface structure impacts particle
interactions with the surface and therefore uptake, which would be described by the interception efficiency for our
measured size range. We account for this factor by increasing the scaling constant for interception in the Emerson et
al. (2020) model. This enhancement resolves the wintertime model-measurement discrepancy in deposition velocity
and suggests further work into seasonal shifts in plant physiology of evergreen needleleaf trees is warranted.
Overall, these findings support the development and addition of seasonal constants into currently used deposition
modules in order to more accurately predict variation in deposition trends, particle lifetime, and the impact of
particles on both air quality and radiative properties.

**Appendix A: Outline of Deposition Model Frameworks and Parameterizations**

We used two main models to explore the roles and possible seasonal changes of deposition mechanisms. The first
was a resistance model developed by Emerson et al. (2020), which is based on the models presented by Zhang et al.
(2001) and Slinn (1982). This model considered four main deposition mechanisms: Brownian motion, gravitational
settling, interception, and impaction. Deposition velocity in the Emerson et al. (2020) model is defined using the
gravitational settling velocity ($V_g$), the aerodynamic resistance ($R_a$), and the surface resistance ($R_s$).

$$V_d(d_p) = V_g(d_p) + \frac{1}{R_a + R_s} \tag{A1}$$

$$V_g(d_p) = \frac{d_p^2 \, \rho_p \, g \, C_c}{18 \, \eta} \tag{A2}$$


$$R_a = \frac{\ln\left(\frac{z_r}{z_0}\right) - \psi_H}{\kappa \, u^*} \tag{A3}$$

$$R_s = \frac{1}{\varepsilon_0 \, u^* (E_b + E_{im} + E_{in}) \, R_1} \tag{A4}$$

$$R_1 = e^{-St^{1/2}} \tag{A5}$$

$$E_b = C_b Sc^{-2/3} \tag{A6}$$





$$E_{im} = C_{im} \left(\frac{St}{\alpha+St}\right)^{\beta} \tag{A7}$$

$$E_{in} = C_{in} \left(\frac{d_p}{A}\right)^{\upsilon} \tag{A8}$$

In these equations $d_p$ is the particle diameter, $\rho_p$ is the density of the particle, $g$ is the gravitational acceleration, $C_c$ is the Cunningham slip correction factor, $\eta$ is the kinematic viscosity of air, $z_r$ is measurement height, $z_0$ is the roughness length, $\psi_H$ is the stability function, $St$ is the stokes number, and $Sc$ is the Schmidt number. The set variables are $C_b = 0.2$, $C_{im} = 0.4$, $\beta = 1.7$, $C_{in} = 2.5$, and $\upsilon = 0.8$ while the variables $\alpha$ and $A$ are land use dependent
and can be found in the original Zhang et al. (2001) publication.

The second model that we explored was from Katul et al. (2010), who presented a multi-layer model that considered the mechanisms of Brownian motion, gravitational settling, impaction, and turbophoresis. This model defined the quasi-laminar boundary layer resistance, or surface resistance, according to the equations outlined by Seinfeld and Pandis (1998) and added an additional term describing turbophoresis.

$$R_s = \frac{1}{\sqrt{-\overline{u'w'(z)}}\left(\theta Sc^{-2/3} + 10^{-3/St_t} + V_t\right)} \tag{A9}$$

Here $\overline{u'w'(z)}$ is the vertical momentum flux or turbulent stress, $\theta$ describes the ratio of the viscous and drag coefficient of the leaf, $St_t$ is the turbulent Stokes number, and $V_t$ is the turbophoretic velocity.

$$\theta = \frac{\pi}{2}\left(\frac{c_v}{c_d}\right) \tag{A10}$$

$$St_t(z) = \frac{V_g\left(-\overline{u'w'(z)}\right)}{g\,\eta} \tag{A11}$$

$$V_t(z) = \frac{\tau_p}{1+\frac{\tau_p}{\tau(z)}}\left(\frac{\sigma_w^2(z)}{\pi\,b\,\eta}\right) \tag{A12}$$

$$\tau_p = \frac{d_p^2\,\rho_p\,C_c}{18\,\eta} \tag{A13}$$

$$\tau = \frac{K_t}{\sigma_w^2} \tag{A14}$$

$$K_t = \frac{-\overline{u'w'}}{\left|\frac{\delta\overline{U}}{\delta z}\right|} \tag{A15}$$

The variable notation follows those described above. The additional variables are $\tau_p$ which is the particle relaxation
time, $\tau$ is the Lagrangian turbulent timescale, $\sigma_w$ is the standard deviation of the vertical velocity, and b which is the thickness of the viscous sublayer for the vertical velocity variance. Katul et al. (2010) used $b = 25$ m and defined an acceptable range for the value as $5 < b < 50$ m. For the calculation of the Lagrangian turbulent timescale, $K_t$ is the eddy viscosity of the flow and is calculated using the vertical momentum flux and the mean longitudinal velocity $(\overline{U})$. These terms are either in relation to or integrated over the height of the canopy $(z)$. It should be noted that for all
equations outlined here, some of the variable notation has been changed from the original papers in order to have consistent references in this work.

**Appendix B: Equations and Results for Leaf Level Energy Balance and Thermophoretic Settling Velocity**

Both real time data and parameters derived from other literature were used to create the leaf level energy balance
equation used in this work. Real time measurements of photosynthetically active radiation ($PAR$), air temperature ($T_{air}$), wind speed ($WS$), and relative humidity ($RH$) were used (Frank et al., 2021). The PAR was converted into short-wave radiation ($SWR$) using the following conversion from Thimijan and Heins (1983):

$$SWR = \left(\frac{PAR}{4.57}\right) \tag{B1}$$





An average stomatal conductance of $0.08 \pm 0.05$ mol m$^{-2}$ s$^{-1}$ was derived from reported literature values (Harley et al., 2014; Calder et al., 2010; McDowell et al., 2008; Sala et al., 2005; Domec et al., 2004; Skov et al., 2004; Hubbard et al., 1999, 2001; Panek and Goldstein, 2001; Ryan et al., 2000; Zhang et al., 1997; Monson and Grant, 1989). Defined parameters and constants used in the model are listed in **Tables B1** and **B2**. The following series of equations were used to form the leaf level energy balance and predict temperature gradients between the air and collecting surface.

Saturation vapor pressure (kPa):

$$e_{sat} = a \, e^{\frac{b \, T_{air}}{(T_{air}+z)}} \tag{B2}$$

Water Vapor Pressure of the air (kPa):

$$e_a = e_{sat} \left(\frac{RH}{100}\right) \tag{B3}$$

Slope of the e$_{sat}$/T curve (kPa °C$^{-1}$):

$$s = \frac{e_{sat} \, b \, c}{(T_{air}+z)^2} \tag{B4}$$

Water vapor pressure deficit of the air (kPa):

$$VPD = e_{sat} - e_a \tag{B5}$$

Absorbed short-wave radiation (W m$^{-2}$):

$$SWR_{abs} = a_{SWR} \cos(i) \, SWR \tag{B6}$$

Incoming long-wave radiation (W m$^{-2}$):

$$LWR_{in} = 1.31 \left(\frac{10 \, e_a}{T_{air}}\right)^{1/7} \sigma_{SB} \, (T_{air} + 273.15)^4 \tag{B7}$$

Isothermal outgoing long-wave radiation (W m$^{-2}$):

$$LWR_{out,i} = \varepsilon \, \sigma_{SB} \, (T_{air} + 273.15)^4 \tag{B8}$$

Isothermal net radiation (W m$^{-2}$):

$$R_{ni} = SWR_{abs} + LWR_{in} - LWR_{out,i} \tag{B9}$$

Leaf boundary-layer resistance (s m$^{-1}$):

$$r_{bl} = \frac{1}{\left(1.5 \, g_x \frac{(WS)^{Jx}}{d^{1-Jx}}\right)} \tag{B10}$$

Radiative resistance (s m$^{-1}$):

$$r_r = \frac{\rho_{air} \, c_p}{4 \, \varepsilon \, \sigma_{SB} \, (T_{air} + 273.15)^3} \tag{B11}$$

Boundary-layer & radiative resistance (s m$^{-1}$):

$$r_{blr} = \frac{1}{(r_{bl}^{-1} + r_r^{-1})} \tag{B12}$$

Modified psychrometric constant (kPa K$^{-1}$):

$$y_m = y \left(\frac{r_{st}}{r_{blr}}\right) \tag{B13}$$





Leaf-to-air temperature difference (C):

725

$$\Delta T = \left( \frac{y_m\, R_{ni}\, r_{blr}}{\rho_{air}\, C_p - VPD} \right) \frac{1}{(s + y_m)}$$

(B14)

Leaf temperature (C):

$$T_{leaf} = T_{air} + \Delta T$$

(B15)

The resulting diel modeled gradients for the winter measurement period are shown in **Figure B1**.

**Table B1**: Leaf level parameters defined for the energy balance.

| Parameter | Units | Value |
|---|---|---|
| Angle from horizontal ($i$) | degrees | $0 - 90$ |
| Absorptance to SWR ($a_{SWR}$) | % | $0.4 - 0.6$ |
| Emissivity ($\varepsilon$) | none | $0.96 - 0.98$ |
| Characteristic dimension ($d$) | mm | 1 * |
| Shape factor of the leaf (shape) | none | 2 ** |
| Stomatal resistance ($r_{st}$) | s m$^{-1}$ | 11.76*** |

*Defined for a pine needle
**Shape = 2 indicates a cylindrical shape
***Value obtained through synthesis of various reported values in the literature for ponderosa pines under normal, unstressed conditions.

730

**Table B2**: Constants used in the formulation of the leaf level energy balance, defined in the order in which they appear in Table S3.

| Constant | Units | Value |
|---|---|---|
| Coefficient in $e_{sat}$ equation ($a$) | kPa | 0.61121 |
| Coefficient in $e_{sat}$ equation ($b$) | none | 17.502 |
| Coefficient in $e_{sat}$ equation ($z$) | °C | 240.97 |
| Stefan-Boltzman constant ($\sigma_{SB}$) | W m$^{-2}$ K$^{-4}$ | 5.67 x 10$^{-8}$ |
| Coefficient in $r_{bl}$ equation for a flat leaf shape ($g_{flat}$) | m | 0.00662 |
| Coefficient in $r_{bl}$ equation for a cylinder leaf shape ($g_{cyl}$) | m | 0.00403 |
| Coefficient in $r_{bl}$ equation for a sphere leaf shape ($g_{sph}$) | m | 0.00571 |
| Coefficient in $r_{bl}$ equation for a flat leaf shape ($j_{flat}$) | none | 0.5 |
| Coefficient in $r_{bl}$ equation for a cylinder leaf shape ($j_{cyl}$) | none | 0.6 |
| Coefficient in $r_{bl}$ equation for a sphere leaf shape ($j_{sph}$) | none | 0.6 |
| Density of air ($\rho_{air}$) | kg m$^{-3}$ | 1.292 |
| Heat capacity of air ($C_p$) | J kg$^{-1}$ K$^{-1}$ | 1010 |
| Psychrometric constant ($y$) | kPa K$^{-1}$ | 0.066 |
| Latent heat of vaporization ($L$) | J g$^{-1}$ | 2500 |



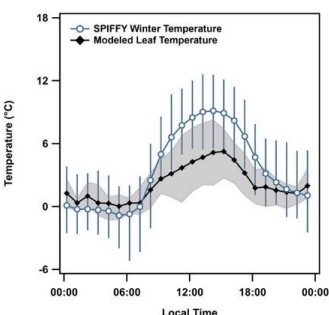

**Figure B1**: Diel average measured air temperature and modeled leaf temperature. With the interquartile range for the measured data shown in bars and the interquartile range for the modeled data shown in shading.

Thermophoretic settling velocity was calculated according to the following equations outlined by Salthammer et al. (2011) and Hinds (1999). The velocity was calculated as:

$$V_{TH} = \frac{3 \, v \, C_c \, H \, \Delta T}{2 \, \rho_{air} \, T_{air}} \tag{B16}$$

using the viscosity of air ($v$), Cunningham slip correction ($C_c$), temperature gradient between the collecting surface and air ($\Delta T$), density of air ($\rho_{air}$), and temperature of air ($T_{air}$). The $H$ term was separately calculated by:

$$H = \frac{1}{1+6\left(\frac{\lambda}{D_a}\right)} \times \frac{\left(\frac{k_a}{k_p}\right)+4.4\left(\frac{\lambda}{D_a}\right)}{1+ 2\left(\frac{k_a}{k_p}\right)+8.8\left(\frac{\lambda}{D_a}\right)} \tag{B17}$$

using the mean free path ($\lambda$), aerodynamic diameter of the particle ($D_a$), the thermal conductivity of air ($k_a$), and the thermal conductivity of the particles ($k_p$). The behavior of the thermophoretic settling velocity as the magnitude of the temperature gradient changes and the thermal conductivity of the particles change is shown in **Figure B2**, along with the results of its integration into the Emerson et al. (2020) model.

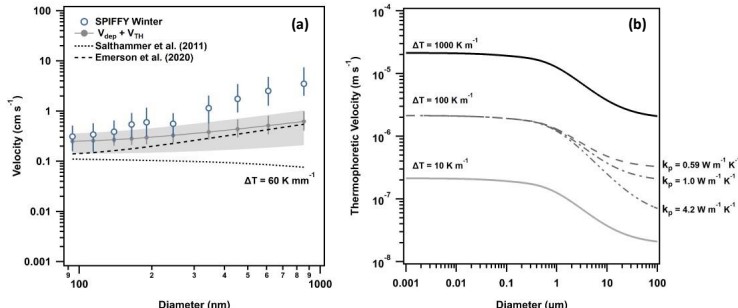

**Figure B2**: (**a**) Integration of the thermophoretic velocity into the Emerson et al. (2020) model. The total modeled velocity is shown in grey with shading representing the standard deviation and bars representing the interquartile range. (**b**) Thermophoretic settling velocity as a function of particle diameter for three different gradient values: $\Delta T$ = 10, 100, and 1000 K m$^{-1}$. Additionally, the velocity at $\Delta T$ = 100 K m$^{-1}$ was varied by particle thermal conductivity and results are shown for $k_p$ = 0.59, 1.0, and 4.2 W m$^{-1}$ K$^{-1}$.

*Data availability*

Data is available at https://manitou.acom.ucar.edu/ with other data from the campaign.



### Author contribution

EKB preformed the analysis presented in this work and prepared the manuscript with contributions from all co-authors. HMD and RF collected the measurements during the SPiFFY campaign in 2016 and provided critical insight into the finer details of the dataset. EWE developed the code used analyze the data. DFK planned the campaign and guided the research. All authors reviewed and edited the manuscript.

### Acknowledgements

We thank Steve Alton of the US Forest Service and the National Center for Atmospheric Research for field site support. We thank John Ortega and Daniel Ziskin for help making the data available through the NCAR MEFO data repository.

### Financial support

Funding for the research was provided by NOAA (Grant NA14OAR4310141) and the Department of Energy (DE-SC0020075) however the publication was not reviewed, or its views endorsed by either organization.

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
