# Peer review of "Seasonal variation in size-resolved particle deposition and the effect of environmental conditions on dry deposition in a pine forest"

_Atmospheric Chemistry and Physics, 2022_

## Referee Comment (RC1)

Review of Seasonal variation in size-resolved particle deposition and the effect of environmental conditions on dry deposition in a pine forest

General comments:

This research examines the seasonality of aerosol dry deposition velocities to an evergreen needleleaf forest in Colorado and attempts to explain seasonal differences through model sensitivity studies where various process representations are tried.  The field experiment is well described, and the model sensitives are methodically presented.  The main premise is that the measurements show significantly greater deposition velocities in the winter than the other seasons.  This conclusion relies on considering only negative (downward) fluxes in the dry deposition velocities.  This practice leads to greater Vd for winter than summer even though the net exchange velocity ($V_{ex}$) is much greater negative (downward) in the summer and the $V_{ex}$ in winter is mostly positive.  I'm not convinced that only counting the negative fluxes is reasonable.  It seems to me that both upward (emission) and downward (deposition) fluxes could be happening simultaneously resulting in a small net $V_{ex}$ that could be either positive or negative.  Negative net flux only means that the average deposition flux is greater than the emission flux for that 30 min period, not that only deposition is occurring.  This needs to be further explained and justified.

The modeling experiments designed to explore possible mechanisms for the greater Vd in winter are methodical and well described.  It is concluded that the only phoretic effect that may be significant in this case is turbophoresis.  These results are interesting and suggest that more models should include this effect.  However, it is also concluded that the effects of turbophoresis are not sufficient to account for the higher winter Vd.  To fully account for the higher winter Vd it was found that increasing the interception scaling coefficient for winter did the trick.  The explanation is that microroughness on the needle surfaces is greater in the winter.  This conclusion takes it for granted that interception is the key process.  This view seems to be based on the apparent success of the Emerson et al (2020) model in better matching observation in forests.  But it should be acknowledged that this model was developed through iterative tuning of empirical constants to observations.  Also, interception is generally considered to be the least physically based process of the collection efficiencies.  For example, in the supplement to the Emerson et al (2020) paper it is said: "There is no underlying physical basis for this term".  After presenting the detailed theory behind the thermophoretic and turbophoretic effects, the tweaking of the interception term does not seem to have the same level of rigor.  There may be other ways to modify the other collection terms to get the desired result.

Specific comments

Lns 157-159:  As mentioned above, I think the assumption of separating positive and negative fluxes should be better explained and justified.

Line 238:  Can you provide more explanation of the uncertainty from counting and implications for the overall uncertainty of the flux measurements?

Line 269:  What are the units for the stability values?

Line 290, eqn 13:  Aerosol dry deposition is better represented by $V_d = \dfrac{V_g}{1-exp(-V_g(R_a+R_s))}$ from Venkatram and Pleim (1999). Eqn 13 overestimates Vd for dp > ~ 8 μm.

Line 299: b is a nondimensional parameter.  There seems to be some confusion about this.  See more in my comment below about line 469

Line 303: This expression makes no sense.  This says that $\sigma_w$ = -0.2$u_*$

Line 372: What is meant by this statement about bi-modal distributions?

Line 425:  Why were the needles consistently colder that the air during the day?  This seems counter intuitive especially if some needles were sunlit.

Line 451:  The role of interception should not be stated as fact.

Line 469:  There is a misunderstanding of the variable *b* from the Katul et al (2010) paper.  This is a non-dimensional parameter.  The viscous sublayer thickness is represented by $\delta$ in that paper and has values on the order of 0.1 – 0.5 mm.  Please correct this.

Line 472: "in" is repeated

Lins 474-478:  The plateau for dp > 10 $\mu$m is not the same as the plateau noted by Saylor et al (2019) in 1-10 $\mu$m range.

Figure 7:  plots are hard to read.

Line 511: the macroscale roughness length used in these models only affects Ra.

Lines 554-558:  Figs S14 and S15 seem to be swapped.

Line 578-579:  The statement that impaction only depends on energy of the particle is not true. It also depends on the obstacle length scale used in the Stoke number.

Line 659:  Expression for St should be given.

Lines 681-682:  Here b is given units of m!  See comment above

Eqn B1:  This expression implies that SWR is smaller than PAR!  They must have different units.

Line 695: stomatal conductance should have units of s/m

Fig B1: How can the leaves be so much colder than the air during the day?

Fig B2:  Units of $\Delta$T are given at K/m.  Isn't $\Delta$T a difference between leaf and air, not a gradient?  Isn't even 10 K/m a ridiculously large gradient?

---

## Referee Comment (RC2)

**Review of "Seasonal variation in size-resolved particle deposition and the effect of environmental conditions on dry deposition in a pine forest"**

The authors present a study focusing on seasonal variation in size-resolved particle deposition velocities above a pine forest in central Colorado. Therefore, particle number flux measurements from each of the four seasons were compared to different model combinations in order to explain the increase in deposition velocity in winter. The authors demonstrated that turbophoresis is an important process which should be integrated into the model for a better deposition velocity estimation. However, this process does not completely fill the gap between measurement and model results. This problem is solved by increasing the interception scaling coefficient in the Emerson et al. (2020) model. The authors hypothesize that the interception scaling coefficient needs to be increased accounting for physiological properties of needles in winter time.

The manuscript is well structured and the different sections well described. I recommend the publication of the manuscript after a few minor corrections.

**Specific comments:**

L17 (Abstract): "Particle concentrations and therefore fluxes were highest ... " Fluxes and concentrations are related, but they do not necessarily react in the same way. Therefore, I would delete the word "therefore" in the sentence. (Same in L47)

L99 – L108: Although the site is described in detail by Ortega et al. (2014), it would be nice to get some information about the measurement height of the different variables. Are all variables measured at the same height?

Figure S2: Why roughness length varies with an amplitude of a few meters during the day? In L 514 you described that the roughness length of a needle leaf forest ranges from 0.8 to 0.9 m. How does this fit together? In addition, assuming that the roughness length is 0.1 times the canopy height, this means that the height of the surrounding trees varies between 10 and 40 meters. Is this correct? Furthermore, I assume that a diurnal variation in roughness length can only occur if the surrounding canopy height differs with wind direction and certain times of day are associated with certain wind directions.

L131: Were particle losses corrected with the method described by Von der Weiden et al. (2009) or was the correction neglected due to the small influence?

Table 1: What does daytime mean? 10:00 to 16:00 local time?

Figure S4: Maybe another color than red for the filtered data points would be better. In a black/white print, the red dots are not really different from the black ones.

L269: Do you mean with "stability" the stability parameter ((z-d)/L)?

L324: Please be careful with the word "largest" because of the negative sign of deposition fluxes. The particle number fluxes are, in the strict sense, smallest and not largest in summer and spring.

L377-L378: What do you mean with "stability function"? Do you mean the stability parameter?

L481/Figure 8: Are the literature values modeled or from measurements?

L583 – L586: What do you think why the turbophoretic included model (Emerson et al. (2020) +  $E_{turbo}$ ,  $C_{in}$ =5) shows a worse agreement than the isolated increased interception model results (Emerson et al. (2020),  $C_{in}$ =5)? Since turbophoresis also influences dry deposition with the same tendency (cf. chapter 3.3.2), I would expect the opposite here.

Technical corrections:

L62: "size resolved" (hyphen is missing)

Figure S1: Please add explanations of the abbreviations (e.g. MFC and UHSAS) to the figure caption.

L153, L197, L214, L221, L231, L377: windspeed  $\rightarrow$  wind speed (with space character)

Table 3: Please add the explanation of the abbreviation LOD. The explanation in the text follows later on the next page.

L333: 129  $\pm$  8 instead of 128  $\pm$  8 nm for the winter median diameters (cf. Table S1).

Table S1 and S2: Please add the unit of the numbers to the table (e.g. in table caption).

L555: I think you mean "µm" instead of "um"?!

Figure S14 (right): I think the unit °C in the figure legend for  $\gamma$  is not correct (contradictory to the text in L545/546).

---

## Author Comment (AC1)

**Response to Reviewer #2**

We thank the reviewer for their comments and feedback and for taking the time to help improve this manuscript. We first respond to the overall comments provided by the reviewer and then list specific responses to line items. Reviewer comments are presented in red and responses in black text.

The main premise is that the measurements show significantly greater deposition velocities in the winter than the other seasons. This conclusion relies on considering only negative (downward) fluxes in the dry deposition velocities. This practice leads to greater Vd for winter than summer even though the net exchange velocity (Vex) is much greater negative (downward) in the summer and the Vex in winter is mostly positive. I'm not convinced that only counting the negative fluxes is reasonable. It seems to me that both upward (emission) and downward (deposition) fluxes could be happening simultaneously resulting in a small net Vex that could be either positive or negative. Negative net flux only means that the average deposition flux is greater than the emission flux for that 30 min period, not that only deposition is occurring. This needs to be further explained and justified.

In response to the need for further justification for the separation of upward and downward fluxes we would like to present three main contentions.

First, we'd like to address the reviewers concerns about positive and negative fluxes happening simultaneously. If this is the case (which it likely is in many of the 30-minute measurement periods) then separating the periods still makes sense. If the goal is to investigate rate of deposition, then a period where emission is clearly dominating and creating a positive flux shouldn't be included in the deposition analysis. So, while it may not be a perfect isolation of deposition it's at least a strong focus on periods where the rate of deposition is dominating which allows us to learn about that process. The fact that this is also an investigation into a rate is important here, because that rate shouldn't change just because the two processes are occurring simultaneously. This means that in a negative flux period where deposition is dominating (but not necessarily occurring alone) we are still able to effectively investigate that rate.

Second, there has been extensive work in the field to identify the drivers of positive and negative fluxes in forested environments. In the text we cite work done by Pryor et al. (2013), as it was done some years before but at the same site where our measurements were done (this has been clarified in the text). This work provides clear evidence for positive fluxes having separate drivers in Manitou Experimental Forest Observatory. If we believe that these are therefore separate processes, then they shouldn't be grouped together to represent deposition.

Finally, we'd like to acknowledge that either separation of upward and downward flux periods or systematic removal periods of positive flux based on other parameters is a common practice in deposition studies. In the text we initially cited Emerson et al. (2020) and Lavi et al. (2013) as these are two recent studies that present only the downward fluxes as deposition. In further support of this practice, we have cited six more deposition studies: Wesely et al. (1985), Beswick et al. (1991), Gallager et al. (1997), Nilsson and Rannik (2001), Vong et al. (2004), and Pryor (2006).

To fully account for the higher winter Vd it was found that increasing the interception scaling coefficient for winter did the trick. The explanation is that microroughness on the needle surfaces is greater in the winter. This conclusion takes it for granted that interception is the key process. This view seems to be based on the apparent success of the Emerson et al (2020) model in better matching observation in forests.

But it should be acknowledged that this model was developed through iterative tuning of empirical constants to observations.

A statement has been made in line 287 to address the iterative tuning of the empirical constants:

"Emerson et al. (2020) iteratively tuned the original empirical constants form Zhang et al. (2001) to expanded sets of measurements over grasslands and forests, so as to better fit observed trends."

Also, interception is generally considered to be the least physically based process of the collection efficiencies. For example, in the supplement to the Emerson et al (2020) paper it is said: "There is no underlying physical basis for this term". After presenting the detailed theory behind the thermophoretic and turbophoretic effects, the tweaking of the interception term does not seem to have the same level of rigor. There may be other ways to modify the other collection terms to get the desired result.

We acknowledge that this may not be the only mechanism responsible for the observations. Although its worth noting that the mechanism would have to have a similar size-dependent trend as the current parameterization of interception. For this reason, we have changed the discussion in the conclusion section of the manuscript to present interception as a hypothesis and not a final answer and to acknowledge that further investigation into deposition mechanisms, including interception, is needed to resolve these kinds of seasonal observations (Line 655ish):

"We account for this factor by increasing the scaling constant for interception in the Emerson et al. (2020) model. This enhancement resolves the wintertime model-measurement discrepancy in deposition velocity and suggests further work into seasonal shifts in plant physiology of evergreen needleleaf trees is warranted. However, it should also be noted that while scaling interception was found to resolve seasonal differences in this work, we acknowledge that this could also be the result of another mechanism with a similar size dependence that is not currently accounted for. Further investigation into the physical basis of interception and other possible mechanisms is still needed."

**Response to Specific Comments:**

Lines 157-159: As mentioned above, I think the assumption of separating positive and negative fluxes should be better explained and justified.

See comments above.

Line 238: Can you provide more explanation of the uncertainty from counting and implications for the overall uncertainty of the flux measurements?

We have provided further contextualization of this error by propagating the flux uncertainty from counting into error for our deposition velocities, as this measurement was the focus of this work. We use the winter and summer periods, as they were the periods of highest and lowest concentrations, showing that lower deposition velocities that occur at night should be treated more carefully as they are on the same order of magnitude as the error, however, this is not problematic for the higher overall deposition during the day:

"The instrumental and random noise were both within the measured variation of the particle fluxes, however, the uncertainty from counting discrete particles exceeded the measured variation. Additionally, the counting error was on average $30 \pm 20$ times larger than the other sources of error. This indicates that the uncertainty from counting is the overwhelming contributor to uncertainty in these flux measurements.

For the winter and summer measurements, during which particle concentrations were lowest and highest, this counting uncertainty in the flux translates to an average error of $7 \pm 4$ cm s-1 and $2 \pm 1$ cm s-1 for the total deposition velocity. Nighttime deposition measurements should therefore be considered less trustworthy, as they are often on the same order of magnitude as this error, however, this is not problematic for the higher deposition velocities measured during the daytime."

Line 269: What are the units for the stability values?

The stability parameter is unitless, calculated as z/L where z is the measurement height above the canopy (m) and L is the Monin-Obukhov length (m). We have rephrased the sentence to make it clearer that we are listing the atmospheric stability parameters for the four periods:

"Atmospheric stability parameters during the periods were -0.03, -0.04, -0.10, and -0.04 for the winter, spring, summer, and fall."

Line 290, eqn 13: Aerosol dry deposition is better represented by $Vd = Vg/(1-exp(-Vg(Ra+Rs)))$ from Venkatram and Pleim (1999). Eqn 13 overestimates Vd for dp > ~ 8 μm.

We acknowledge that Eq. 13 is not entirely consistent with mass conservation and therefore has some limitations. However, this parameterization is still being widely used in many global models and is able to predict size-dependent trends, especially in the accumulation mode which is the size range of focus in this work. A statement has been added to the methods section, which parallels our statement in Line 376, to clarify that other viable parameterizations of particle deposition exist, however, this model was chosen because of its wide use in global models and ease of use for assessing the influence of underlying mechanisms on overall deposition:

"While other viable parameterizations of particle deposition exist (Sehmel, 1973; Kramm et al., 1992; Venkatram and Pleim, 1999), the parameterization presented by Emerson et al. (2020), Zhang et al. (2001), and Slinn (1982) was chosen for this work because of its wide use and its ability to assess the influence of individual mechanisms to overall deposition."

In this statement we have added references to Venkatram and Pleim 1999, as well as Sehmel (1973) and Kramm et al. (1992).

Line 299: b is a nondimensional parameter. There seems to be some confusion about this. See more in my comment below about line 469

This has been corrected, see full response bellow.

Line 303: This expression makes no sense. This says that σw = -0.2u*

Thank you for pointing out that this notation creates a non-sensical statement. We have changed this to "σw ≈ 1.0u* to 1.2u*" to better indicate that this relationship exists in the range of 1.0u* to 1.2u*.

Line 372: What is meant by this statement about bi-modal distributions?

This statement was in reference to the conclusions of Mammarella et al. (2011), where it was concluded that the observed increase in total winter deposition was due to an increase in concentration of larger

particles in the winter distribution that was not present in the other seasons. The increase in the larger mode was stated to drive overall deposition up in the winter. We have rephrased to better reflect this:

"That work concluded that the observed seasonal dependence of the dry deposition was driven by changes in particle distributions, with higher concentrations of larger particles in the wintertime."

Line 425: Why were the needles consistently colder that the air during the day? This seems counter intuitive especially if some needles were sunlit.

So, there are two important pieces in answering this question: (1) the influence of other physiological processes on plant temperature and (2) the limitations of the model that we used to estimate leaf temperature.

Leaves can be cooler than the reported air temperature in the day due even if they are sunlit, a great example of this is when evaporative cooling is taking place at the surface of the leaf. So the needle being colder during the day is not outside of the realm of possibilities.

That being said, it's more likely that this consistent lower temperature is caused by using a single value of estimated stomatal conductance and resistance from other literature. We present a prediction that was based on an estimated minimum stomatal resistance from the $0.08 \pm 0.05$ mol m-2 s-1 conductance. If we include the range of possible leaf temperatures based on the range of resistances (derived from the literature values and then averaged) we can present this range in calculated leaf temperatures:

[Figure]

This is the average bounds of calculated temperature using a stomatal resistance range of 1 s/m to 45 s/m.

Even with the range we end up with average temperature differences of $7 \pm 3$ C (1 s/m) to $1 \pm 1$ C (45 s/m), which combined give an average difference of $4 \pm 4$ C.

In order to be more transparent about the possible range of estimated leaf temperatures figure B2 has been replaced with the one above and a sentence has been included in the thermophoresis discussion:

"We modeled these gradients using measured meteorological parameters along with a simple leaf level energy balance (**Appendix B**). A range of stomatal resistance values were used to bound the predicted leaf temperature (**Figure B1**). The leaf level energy balance predicted an average temperature difference of $4 \pm 4$ K between the leaf and surrounding air."

Line 451: The role of interception should not be stated as fact.

We have rephrased this sentence to state:

"First, the model neglects interception, which is believed to have a significant role in deposition over forests."

Line 469: There is a misunderstanding of the variable b from the Katul et al (2010) paper. This is a non-dimensional parameter. The viscous sublayer thickness is represented by δ in that paper and has values on the order of 0.1 – 0.5 mm. Please correct this.

Thank you for helping to clarify this, I see that this was a mistake in interpreting the relationship listed between δ and the non-dimensional b parameter in the original text. The reference to b has been changed in the text:

Line 301: "… a non-dimensional variable (b), which can range from $5 < b < 50$, …"

Line 472: "As a sensitivity test, we varied the non-dimensional parameter (b) between 5 and 50 following Katul et al. (2010)"

Figure 7: "The shaded range shows how the second model formulation changes with the non-dimensional parameter ($b$)…"

Line 689: "… and $b$ which is  a non-dimensional parameter. Katul et al. (2010) used $b = 25$ and defined an acceptable range for the value as $5 < b < 50$."

Line 472: "in" is repeated

Thank you for catching this, the second instance of "in" has been removed

Lins 474-478: The plateau for dp > 10 μm is not the same as the plateau noted by Saylor et al (2019) in 1-10 μm range.

You are correct. This was not to say that the two were the same only to point out that incorporation of turbophoresis created the same kind of plateau in the trend.

There is currently no physical basis of understanding for the appearance of this plateau in the data, and Saylor et al (2019) acknowledges this. Their solution was a purely empirical fit to the data, which was then attributed to an unknown mechanism. Here we hypothesize that turbophoresis could potentially be a contributor to this trend given that it creates the same kind of plateau seen in the data (albeit not in the correct size range with the current expression used in this work). However, we also acknowledge that more work directly targeting the mathematical representation of turbophoresis would be needed to confirm this. The concluding statements in this paragraph have been edited to more explicitly state this:

"Turbophoresis thus provides a potential mechanism to explain this observed feature in $1 – 10$ μm size range as it causes a similar feature in the size-dependent trend. However, the added plateau from turbophoresis does not fall in correct size range. A more critical exploration of the turbophoresis parameterization is therefore needed to resolve whether it is a viable explanation for the observed trend."

Figure 7: plots are hard to read.

In Figure 7 legends and axis text has been increased:

[Figure]

Line 511: the macroscale roughness length used in these models only affects Ra.

That is correct, we've made a note that explicitly states this:

"This roughness length is used in the calculation of the aerodynamic resistance above the canopy."

Lines 554-558: Figs S14 and S15 seem to be swapped.

The two have been swapped in the supplemental to reflect their reference in the main text.

Line 578-579: The statement that impaction only depends on energy of the particle is not true. It also depends on the obstacle length scale used in the Stoke number.

That is correct, our intention in this sentence was not to state that it was only dependent on the energy of the particle. This was more a way to contrast it to what is traditionally considered interception and why the discussed changes to the leaf surface would impact interception over impaction. Both interception and impaction depend on the needle length and therefore surface available for depositing particles, but while impaction is dependent on the particle's momentum/energy interception is not currently tied to that and so changes to surface structure that are changing microscale roughness and surface-particle interactions would have a larger impact on interception. An amendment has been made to the concluding sentence so as to remove the idea that Brownian motion and impaction are only dependent on particle energy:

"In contrast, mechanisms like Brownian motion and impaction both depend on the energy of the particle along with the size of the collecting surface."

Line 659: Expression for St should be given.

An expression for St has been added to the listed equations.

Lines 681-682: Here b is given units of m! See comment above

Units have been removed see comments above.

Eqn B1: This expression implies that SWR is smaller than PAR! They must have different units.

You are correct, there is a unit conversion built into that equation. I have added a statement so that the unit conversion in this relationship is clear:

"In this conversion, PAR with units of µmol s-1 m-2 is being converted to SWR with units of W m-2 giving the conversion factor units of µmol s-1 m-2 per W m-2."

Line 695: stomatal conductance should have units of s/m

The units of stomatal conductance are mol m-2 s-1, however, an average minimum stomatal resistance estimate has been added which has units of s m-1:

"An average stomatal conductance of $0.08 \pm 0.05$ mol m-2 s-1 was derived from reported literature values, which translated to an estimated minimum stomatal resistance of $23 \pm 22$ s m$^{-1}$ (Harley et al., 2014; Calder et al., 2010; McDowell et al., 2008; Sala et al., 2005; Domec et al., 2004; Skov et al., 2004; Hubbard et al., 1999, 2001; Panek and Goldstein, 2001; Ryan et al., 2000; Zhang et al., 1997; Monson and Grant, 1989)."

Fig B1: How can the leaves be so much colder than the air during the day?

See comments above.

Fig B2: Units of ΔT are given at K/m. Isn't ΔT a difference between leaf and air, not a gradient? Isn't even 10 K/m a ridiculously large gradient?

I see that this could have been made clearer, apologies. The ΔT presented here is in fact a gradient – which is what is required by the parameterization of thermophoretic settling velocity presented by Salthammer et al. (2011). So those units are correct.

In order to differentiate between difference and gradient a ∇ should have been used instead of a Δ, this replacement has been made in the text and in Figure B2. Additionally, in line 431 the term difference has been used to describe ΔT and gradient has been used to describe ∇T.

Also, yes, 10 K/m is a ridiculous gradient, the ones needed here are totally outlandish and completely out of the realm of possibilities.